# Temporally controlled multistep division of DNA droplets for dynamic artificial cells

Tomoya Maruyama [1], Jing Gong[1] & Masahiro Takinoue [1,2,3] ✉

Synthetic droplets mimicking bio-soft matter droplets formed via liquid-liquid phase separation (LLPS) in living cells have recently been employed in nano-biotechnology for artificial cells, molecular robotics, molecular computing, etc. Temporally controlling the dynamics of synthetic droplets is essential for developing such bio-inspired systems because living systems maintain their functions based on the temporally controlled dynamics of biomolecular reactions and assemblies. This paper reports the temporal control of DNA-based LLPS droplets (DNA droplets). We demonstrate the timing-controlled division of DNA droplets via time-delayed division triggers regulated by chemical reactions. Controlling the release order of multiple division triggers results in order control of the multistep droplet division, i.e., pathway-controlled division in a reaction landscape. Finally, we apply the timing-controlled division into a molecular computing element to compare micro-RNA concentrations. We believe that temporal control of DNA droplets will promote the design of dynamic artificial cells/molecular robots and sophisticated biomedical applications.

Living cells exhibit well-organized dynamics in bio-soft matter assemblies, such as membrane deformation, cell division, and cell differentiation[1], which are essential features that distinguish living systems from non-living matter. Recently, liquid-liquid phase separation (LLPS) droplets of bio-soft matter have been found in living cells, and their dynamic behaviors have attracted attention[2,3], such as nucleolar assembly through non-equilibrium processes of rRNA transcription[4], sol-gel transition[5], and activation/inhibition of molecular reactions[6]. These examples show that precise temporal control of biological LLPS droplets via non-equilibrium chemical reactions realizes such dynamic behaviors.

Synthetic LLPS droplets have recently been explored in bottom-up synthetic biology for constructing artificial cells[7,8], molecular robots[9], molecular computers[10,11], and biomedical nanodevices[12]. Various dynamic behaviors of synthetic LLPS droplets have been reported, such as sequestration of molecules[13–15], motion[9,16], and division[17]. More recently, non-equilibrium dynamics such as cyclic assembly/

disassembly[18,19] and transient shell-formation[20] of synthetic coacervate droplets were achieved by coupling LLPS droplets with non-equilibrium chemical reactions such as phosphorylation/dephosphorylation[21,22] and enzymatic synthesis of polynucleotide[23]. However, temporal control of LLPS droplet dynamics remains difficult. Programmable temporal control methods must be developed to mimic cell dynamics.

DNA is well known for its programmable structures[24,25] and reactions[26]. DNA programmability also facilitates the temporal control of chemical reactions. For example, DNA computing reactions have been demonstrated, such as the chemical oscillation of DNA concentrations[27–29], temporal logic circuit[30], and timing-controlled generation of chemical signals[31,32]. Moreover, the programmability of DNA has been utilized not only for controlling chemical reactions but also for controlling the physical dynamics of mechanical DNA-based nanostructures[33–35]. Particularly, DNA-based coacervates[36,37] (also referred to as DNA droplets) formed with branched DNA

[1]Department of Life Science and Technology, Tokyo Institute of Technology, 4259 Nagatsuta-cho, Midori-ku, Yokohama, Kanagawa 226-8501, Japan. [2]Department of Computer Science, Tokyo Institute of Technology, 4259 Nagatsuta-cho, Midori-ku, Yokohama, Kanagawa 226-8501, Japan. [3]Research Center for Autonomous Systems Materiology (ASMat), Institute of Innovative Research, Tokyo Institute of Technology, 4259 Nagatsuta-cho, Midori-ku, Yokohama, Kanagawa 226-8501, Japan. ✉e-mail: takinoue@c.titech.ac.jp

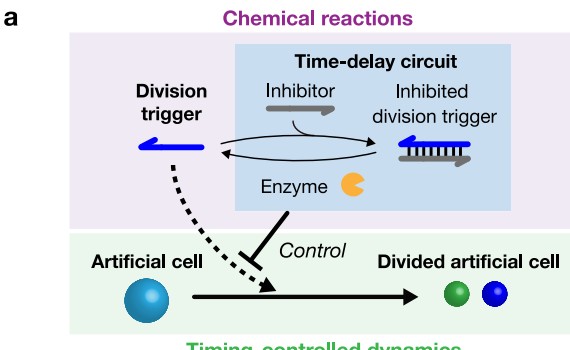

**a** Chemical reactions

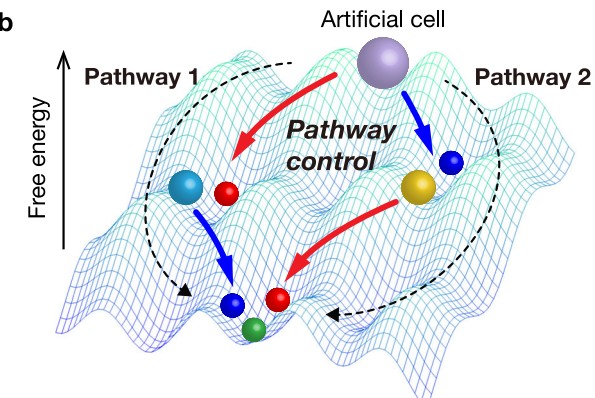

**b**

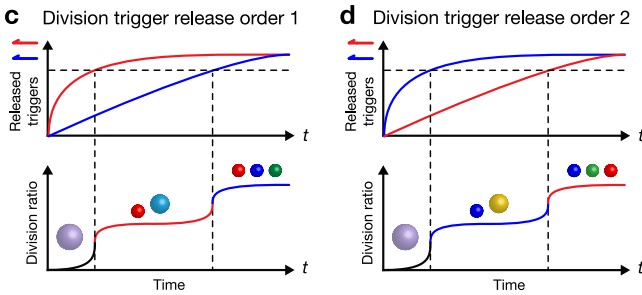

**c** Division trigger release order 1

**d** Division trigger release order 2

**Fig. 1 | Conceptual illustration of timing control of artificial cell division via chemical reactions. a** Timing-controlled division of an artificial cell regulated by a time-delay circuit. **b** Pathway-controlled division of DNA droplet-based artificial cells. **c, d** Time-delay circuits control division pathways by changing the release order of different division triggers.

trigger DNAs. Here, we couple the mixed DNA droplet with chemical reactions; the time-delayed release of division triggers (Fig. 1a) realizes timing and pathway control of DNA droplet division (Figs. 1b–d). We use temporal control of DNA reactions based on RNA degradation with a ribonuclease H (RNase H), which has been used in many dynamic DNA reactions such as DNA oscillators[27], DNA bistable switch[49], logic computation[50], DNA walker[51], and timers for DNA strand displacement reactions[31,32]; however, there is no report on temporal control of LLPS droplets with the RNase H reaction. Finally, we present a molecular computing element to compare the concentrations of microRNA (miRNA) sequences (called molecular comparators) as an application of the timing-controlled division of DNA-droplet-based artificial cells. Our results provide a method for chemically regulating the timing-controlled physical dynamics of LLPS droplets for artificial cell studies.

## Results

### Autonomous division of binary-mixed DNA droplets using division trigger DNAs

Figure 2a shows the design of DNA droplets for artificial cells. Y-shaped branched DNA nanostructures self-assemble to form DNA droplets via hybridization of self-complementary sticky ends at their branches[17]. Because $Y_A$ and $Y_B$ have non-complementary sticky ends (Fig. 2b; detailed sequences are in Supplementary Table 1), the resultant A- and B-droplets do not fuse; however, a 6-branched DNA linker ($L_{AB}$) (Fig. 2b; Supplementary Table 2) can crosslink $Y_A$ and $Y_B$, forming a binary-mixed DNA droplet (A·B-droplet) (Fig. 2b). Here, '·' (a single center dot) in 'A·B' indicates that one type of linker ($L_{AB}$) crosslinks $Y_A$ and $Y_B$ in A·B-droplet. Figure 2c shows confocal laser scanning microscopy (CLSM) images of the A·B-droplet. The A·B-droplet can be divided by cleaving $L_{AB}$ into two portions (Fig. 2d). For $L_{AB}$ cleavage, we used a nucleic acid strand displacement reaction induced by single-stranded DNA (ssDNA) division triggers ($T_{AB1}$ and $T_{AB2}$) (Fig. 2e). This design is based on our previous study[47]. The division triggers hybridize to the toehold sequences ($Toehold_{AB1}$ and $Toehold_{AB2}$) in $L_{AB}$ and invade the branches of $L_{AB}$ via strand-displacement reactions (Fig. 2e, middle), cleaving $L_{AB}$ into two portions (Fig. 2e, right). After adding the division triggers, the cleaved-$L_{AB}$ is more stable than the initial $L_{AB}$ because of the Gibbs free energy change ($\Delta G_{Clv}$) induced by division trigger hybridization and strand displacement reactions, leading to the droplet division (Fig. 2f). Figure 2g shows the time-lapse images of the division of the A·B-droplet after adding the division triggers. The A·B-droplet started to divide just after adding division triggers. The result agreed with the previous study[47]; although a slightly inhomogeneous area richer in $Y_A$ or $Y_B$ component was observed in the A·B-droplet, the inhomogeneity was not necessary for a droplet division.

### Design of timing-control of DNA droplet division based on time-delay circuits

We hypothesized that inhibiting released division triggers causes the time delay of the linker cleavage, resulting in timing control over DNA droplet division. Figure 3a shows the design of a time-delay circuit comprising reactions (i) and (ii). (i) Released division triggers changed to inhibited division triggers by the hybridization of excess single-stranded RNAs (ssRNAs), named inhibitor RNAs. (ii) An RNase H degrades the inhibitor RNAs in the inhibited division triggers, thereby releasing released division triggers. These two reactions cause a time delay in the cleavage of the DNA linker.

To tune the time delay of the binary-mixed DNA droplet division, we introduced $L^\dagger_{AB}$ in addition to the original DNA linker, $L_{AB}$ (Fig. 3b). We describe this binary-mixed DNA droplet as "A:B-droplet," where ':' (double dots) indicates that $Y_A$ and $Y_B$ are crosslinked with two DNA linkers, $L_{AB}$ and $L^\dagger_{AB}$. A:B-droplets divide only when both $L_{AB}$ and $L^\dagger_{AB}$ are cleaved. In addition, linkers and triggers with "†" indicate those that can achieve a time delay in the presence of inhibitor RNAs and RNase H (Fig. 3c). $L_{AB}$ is cleaved by the released division triggers $T_{ABi}$ ($i = 1, 2$),

Now we demonstrate the timing-controlled physical dynamics of DNA droplet-based artificial cells by coupling them with chemical reactions exhibiting a transient non-equilibrium relaxation process, resulting in the pathway control of artificial cell division (Fig. 1). We use DNA droplets constructed by mixing two Y-shaped branched DNA nanostructures ($Y_A$ and $Y_B$; called binary-mixed DNA droplets), in which 6-branched DNA linkers crosslinked $Y_A$ and $Y_B$ (Figs. 2a, b). Mixed DNA droplets are divided into $Y_A$- and $Y_B$-droplets by cleaving the DNA linkers through the hybridization with division

nanostructures[17,38–45] can couple physical dynamics with chemical reactions in a programmable manner. DNA droplets divide autonomously with enzymatic[17] and photo[41] cleavage reactions and locomotion via enzymatic degradation[16,46]. Phase separation of DNA droplets based on molecular logic computation[47] and reaction-diffusion pattern formation coupled with RNA transcription and diffusion[48] have also been demonstrated. However, achieving the timing-controlled physical dynamics of DNA droplets coupled with chemical reactions remains challenging.

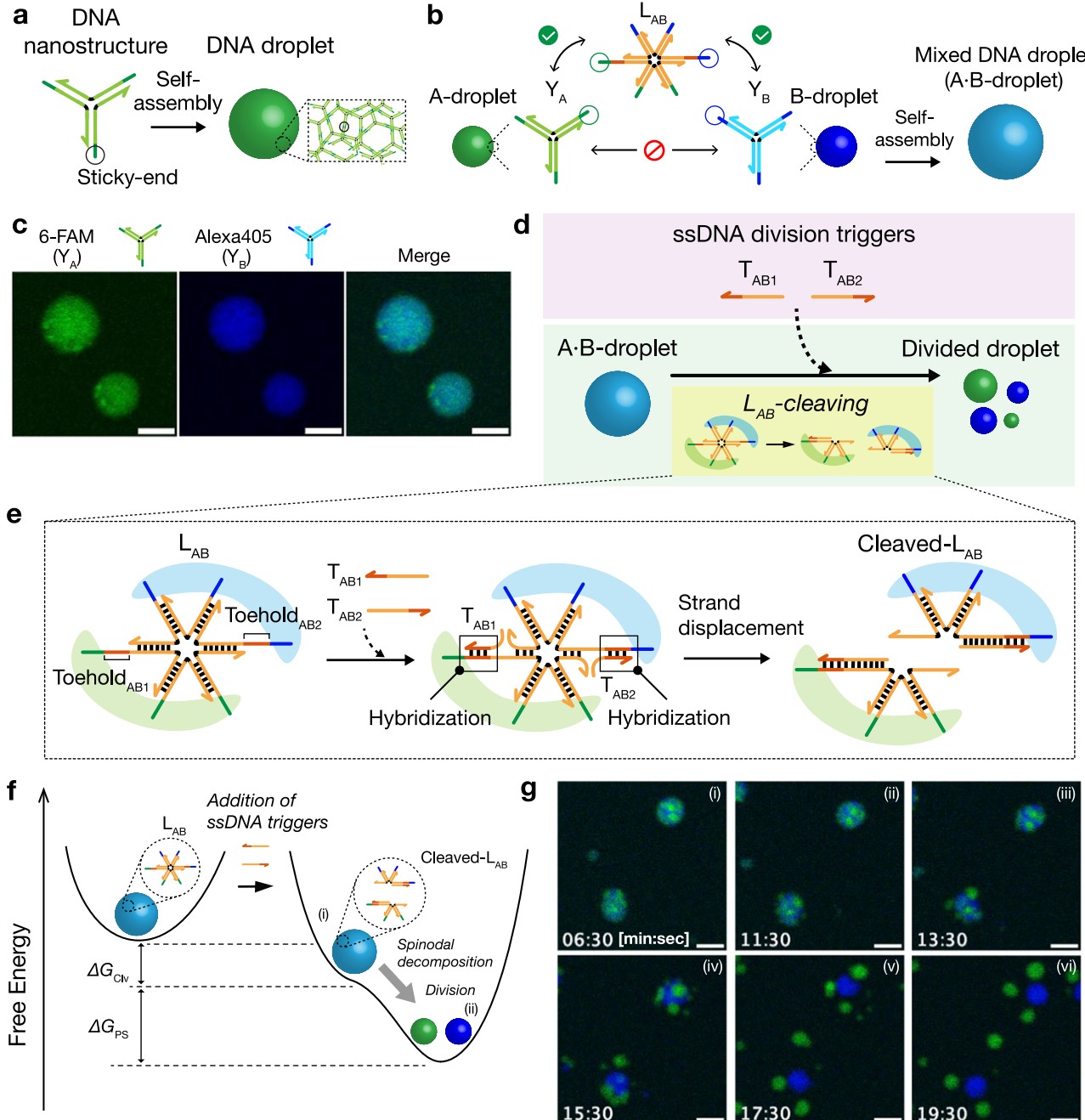

**Fig. 2 | Design of binary-mixed DNA droplets. a** Schematic of DNA droplet formation. Y-shaped branched DNA nanostructures self-assemble via binding of palindromic sticky ends, forming a DNA droplet. **b** Binary-mixed DNA droplet formation. Sticky ends of $Y_A$ and $Y_B$ are crosslinked by 6-branched DNA linker $L_{AB}$. After the self-assembly of these DNA nanostructures, a binary-mixed DNA droplet (A·B-droplet) is formed. **c** CLSM images of A·B-droplets. Green: $Y_A$ labeled with 6-carboxyfluorescein (6-FAM); Blue: $Y_B$ labeled with Alexa Fluor® 405 (Alexa405). Co-localization of $Y_A$ and $Y_B$ was observed. Scale bars: 10 μm. Experiments were repeated three times independently with similar results. **d**, **e** Division of A·B-droplet

via $L_{AB}$ cleavage. $L_{AB}$ is designed to be cleaved by a strand-displacement reaction with ssDNA division triggers ($T_{AB1}$ and $T_{AB2}$). **f** Description of the A·B-droplet division dynamics based on reaction landscapes. The ssDNA division triggers change the reaction landscape from a single-minimum shape: (i) A·B-droplet with ssDNA triggers but the A·B-droplet is not divided yet; (ii) A- and B-droplets are divided through the spinodal decomposition. $\Delta G_{Clv}$ and $\Delta G_{PS}$ are Gibbs free energy changes for the linker cleavage reaction and the phase separation, respectively. **g** Time-lapse images of A·B-droplet division. Scale bars: 10 μm. Experiments were repeated three times independently with similar results.

while $L^{\dagger}_{AB}$ is cleaved by released division triggers $T^{\dagger}_{ABi}$ ($i = 1, 2$). Inhibitor RNAs $R^{\dagger}_{ABi}$ hybridize with $T^{\dagger}_{ABi}$, and form inhibited division triggers $iT^{\dagger}_{ABi}$, inducing the time delay of A:B-droplet division. This time-delay circuit was inspired by intracellular time-delay control via reaction suppression based on small RNA expression[52]. For such biological meaning and applications shown later, we used natural miRNA sequences, miR-6875-5p and miR-4634[47,53], for $R^{\dagger}_{ABi}$ sequences,

respectively (Supplementary Table 3); that is, if either of the miRNAs exist, the A:B-droplet division is delayed.

**Numerical investigations of timing-control of DNA droplet division**

First, we numerically investigated the dependence of the cleaving rate of the DNA linker $L^{\dagger}_{AB}$ on the concentrations of RNase H and the

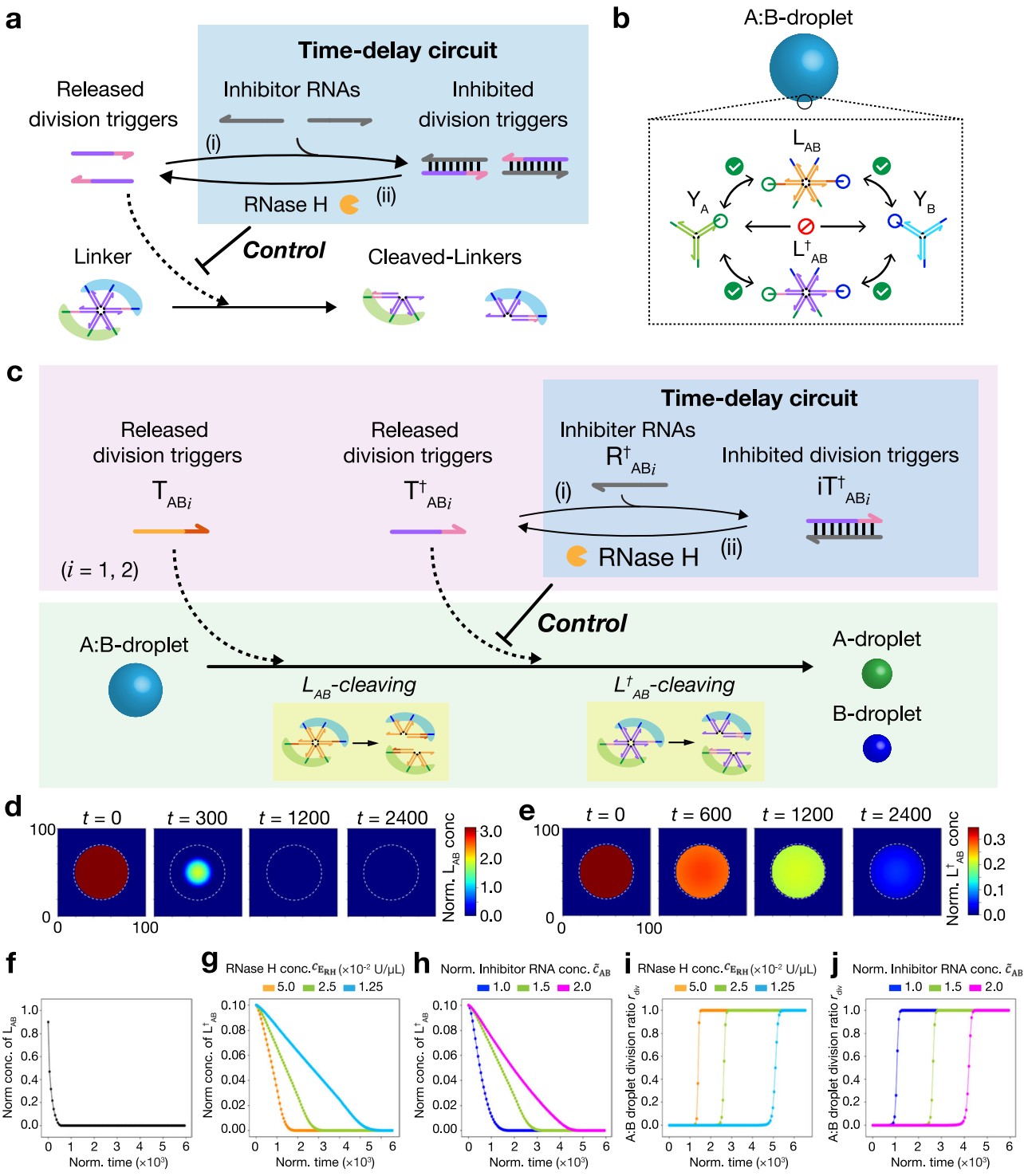

inhibitor RNAs ($R^\dagger_{ABi}$; $i = 1, 2$) when the time-delay circuits work. By assuming that the phase separation dynamics follows the spatiotemporal DNA linker distribution in a mixed DNA droplet, we used a reaction-diffusion model based on the partial differential equations (details in Supplementary Note 1) to estimate the spatiotemporal distribution. The following equations denote the spatiotemporal change of the division triggers $T^\dagger_{ABi}$ ($i = 1, 2$) controlled by the time-delay circuit:

$$\frac{\partial u_{T^\dagger_{ABi}}}{\partial t} = D(\boldsymbol{x})\nabla^2 u_{T^\dagger_{ABi}} - f_{H\text{-}SD}(\boldsymbol{u}) + g_{TD}\left(\boldsymbol{u}; c_{E_{RH}}, u^0_{R^\dagger_{ABi}}\right) \quad (1)$$

$$g_{TD}\left(\boldsymbol{u}; c_{E_{RH}}, u^0_{R^\dagger_{ABi}}\right) := \frac{k_{cat}c_{E_{RH}}u_{iT^\dagger_{ABi}}}{K_m + u_{iT^\dagger_{ABi}}} - k_{h_{RNA}}u_{T^\dagger_{ABi}}u_{R^\dagger_{ABi}} \quad (2)$$

where $u_X$ is the concentration of molecule "X"; $\boldsymbol{u} := \{u_{T^\dagger_{ABi}}, u_{iT^\dagger_{ABi}}, u_{R^\dagger_{ABi}}, ...\}$ is the vector of concentrations of molecules. The first term in Eq. (1) is the spatial diffusion of $T^\dagger_{ABi}$; $D(\boldsymbol{x})$ is the diffusion coefficient depending on the position $\boldsymbol{x}$ ($\boldsymbol{x}$ = "inside" or "outside" of A:B-droplet). The second term $-f_{H\text{-}SD}(\boldsymbol{u})$ denotes the consumption of division triggers $T^\dagger_{ABi}$ via hybridization and strand displacement with the linker $L^\dagger_{AB}$. The third term $g_{TD}\left(\boldsymbol{u}; c_{E_{RH}}, u^0_{R^\dagger_{ABi}}\right)$ denotes the time-

**Fig. 3 | Numerical investigation of timing-controlled linker-cleavage for DNA droplet division. a** Schematics of time-delay circuit to regulate cleaving rate of a DNA linker. (i) Excess inhibitor RNAs hybridize with released division triggers, producing inhibited division triggers. (ii) Released division triggers are released from inhibited division triggers by RNase H reaction. Released division triggers hybridize with the DNA linker, cleaving the linker via strand displacement. **b** Compositions of a binary-mixed DNA droplet (A:B-droplet). **c** Schematic of the timing-controlled division of A:B-droplet using a time-delay circuit. $L_{AB}$ is initially cleaved using $T_{ABi}$ followed by the cleaving of $L^†_{AB}$, resulting in the division of A:B-droplet. The time-delayed cleaving of $L^†_{AB}$ is achieved by the release of $T^†_{ABi}$ from $iT^†_{ABi}$. The degree of time delay of the $T^†_{ABi}$ release decides the timing control of the droplet division. Linkers and triggers with "†" indicate those that can achieve a time-delay circuit if inhibitor RNAs and RNase H are added. **d, e** Snapshots of numerically calculated concentrations of $L_{AB}$ and $L^†_{AB}$ in the A:B-droplet using the reaction-diffusion simulation. The white broken-line circle indicates the surface of the A:B-droplet. **f** Time course of $L_{AB}$ of the A:B-droplet in the numerical simulation. **g, h** Time courses of $L^†_{AB}$ concentrations of the A:B-droplet in the numerical simulation via changing the RNase H concentration $c_{E_{RH}}$ or inhibitor RNA concentration $u^0_{R^†_{ABi}}$, respectively. Normalized initial total concentration of inhibitor RNA is defined as $\tilde{c}_{AB} = u^{tot}_{R^†_{ABi}} / u^{tot}_{T^†_{ABi}}$ $(i = 1,2)$, where $u^{tot}_{R^†_{ABi}} = u^0_{R^†_{ABi}} + u^0_{iT^†_{ABi}}$ is the initial total concentration of excess and hybridized inhibitor RNAs, and $u^{tot}_{T^†_{ABi}} = u^0_{T^†_{ABi}} + u^0_{iT^†_{ABi}}$ is the initial total concentration of released and inhibited triggers. **g** $c_{E_{RH}} = 1.25$, $2.5$, and $5.0 \times 10^{-2}$ U/μL; $\tilde{c}_{AB} = 1.5$. **h** $c_{E_{RH}} = 2.5 \times 10^{-2}$ U/μL; $\tilde{c}_{AB} = 1.0$, $1.5$, and $2.0$. **i, j** Time courses of the division ratio $r_{div}$ in the reaction-diffusion simulation with changing the $c_{E_{RH}}$ or $u^0_{R^†_{ABi}}$, respectively. **i** $c_{E_{RH}} = 1.25$, $2.5$, and $5.0 \times 10^{-2}$ U/μL; $\tilde{c}_{AB} = 1.5$. **j** $c_{E_{RH}} = 2.5 \times 10^{-2}$ U/μL; $\tilde{c}_{AB} = 1.0$, $1.5$, and $2.0$. Source data are provided as a Source Data file.

delay circuit reaction composed of the generation and inhibition of $T^†_{ABi}$, described in Eq. (2) in detail; $K_m$ and $k_{cat}$ are the Michaelis-Menten parameters for the RNase H reaction; $c_{E_{RH}}$ is the total RNase H concentration; $k_{h_{RNA}}$ are the hybridization rates of the division triggers with inhibitor RNAs; $u^0_{R^†_{ABi}}$ $(i = 1, 2)$ are the initial concentrations of excess inhibitor RNAs. Thus, the time course of $T^†_{ABi}$ is controlled by two important factors of the time-delay circuit: $c_{E_{RH}}$ and $u^0_{R^†_{ABi}}$.

Figures 3d and e show the distributions of $L_{AB}$ and $L^†_{AB}$, respectively, in an A:B-droplet at several normalized simulation time steps (the white broken-line circle indicates the surface of the A:B-droplet). In the present study, we fixed the percentages of $L_{AB}$ and $L^†_{AB}$ to the total amount of linker DNA to 90% and 10%, respectively. We referred to previously reported kinetic parameters and diffusion coefficients[50,54–57]. The degradation of $L_{AB}$ occurs from the outside of the droplet towards the inside, while that of $L^†_{AB}$ happens uniformly throughout the droplet. This would be because the reaction rate is faster than the diffusion rate for $L_{AB}$, whereas the reaction rate is slower than the diffusion rate for $L^†_{AB}$ due to the low amount of released division triggers. The results show that $L^†_{AB}$ remains longer than $L_{AB}$, although the percentage of $L^†_{AB}$ is lower than that of $L_{AB}$. This indicates that the decrease of $L^†_{AB}$ becomes slower due to the time-delay circuit.

Figure 3f is the time course of DNA linker $L_{AB}$ cleavage, showing that $L_{AB}$ cleaves rapidly. Next, we investigated the dependence of the cleavage rate of DNA linkers $L^†_{AB}$ on the total RNase H concentration ($c_{E_{RH}}$) and the initial concentration of the excess inhibitor RNAs ($u^0_{R^†_{ABi}}$; $i = 1, 2$) (Figs. 3g and h). The cleavage rate of the $L^†_{AB}$ becomes slower by decreasing RNase H concentration $c_{E_{RH}}$ (Fig. 3g) or increasing RNA concentration $u^0_{R^†_{ABi}}$ (Fig. 3h). By summing them up, the decreasing time courses of total linker concentrations are found to be delayed (Supplementary Fig. 1). Here, we assume that the division ratio of the A:B-droplets $r_{div}$ follows a sigmoidal cooperative function of the total concentration of uncleaved linkers $w$ (Supplementary Note 1):

$$H(w) = \frac{K^n}{K^n + w^n} \quad (3)$$

$$r_{div} = \frac{H(w) - H_{min}}{H_{max} - H_{min}} \quad (4)$$

where $K$ is the threshold concentration of the uncleaved linker for A:B-droplet division and $n$ is a cooperativity coefficient that expresses the switch-like dependence of division on $w$. $H_{max}$ and $H_{min}$ are the maximum and minimum values of $H(w)$, respectively. The cooperative switch-like dependence expressed by the Hill-type function $H(w)$ was observed by Gong et al.[47]; cooperative nonlinear behavior is observed because the cleavage of most linkers is necessary for DNA droplet division. Figures 3i and j show the time courses of $r_{div}$ when changing RNase H concentration $c_{E_{RH}}$, and RNA concentration $u^0_{R^†_{ABi}}$ with $K = 0.05$

and $n = 16$ fixed. Consequently, the $r_{div}$ increases at a slower rate by decreasing $c_{E_{RH}}$ or increasing $u^0_{R^†_{ABi}}$. This trend did not change depending on values of $K$ and $n$ (Supplementary Fig. 2). Therefore, these results suggest that the timing of the division can be controlled by tuning the cleavage rate of $L^†_{AB}$.

## Experimental investigations of timing-control of DNA droplet division

We performed the experiments shown in Fig. 3c for the timing-controlled division of the A:B-droplets. The droplet division reaction started by adding released triggers ($T_{ABi}$), inhibited triggers ($iT^†_{ABi}$), excess inhibitors ($R^†_{ABi}$), and RNase H into an A:B-droplet solution (Methods in detail). Figures 4a and b show time-lapse images of A:B-droplet division. The required time for the division was elongated with decreasing $c_{E_{RH}}$ or increasing $u^0_{R^†_{ABi}}$. Furthermore, we quantified the division ratio $r_{div}$ of the A:B-droplet using image processing (see Supplementary Note 3) (Fig. 4c, d). $r_{div}$ is 0 if the A- and B-droplets are fully mixed in the A:B-droplets, and 1 if the A:B-droplets are completely divided into A- and B-droplets. The results demonstrated that the increasing rate of $r_{div}$ became slower with decreasing $c_{E_{RH}}$ or increasing $u^0_{R^†_{ABi}}$, which is consistent with the numerical simulation results. The time courses of $r_{div}$ in the experiments were not as sharp as those in the simulation, probably because of the slow response of the B-droplet against linker cleavage. From the experimental results, we concluded that the timing-controlled division of DNA droplets was achieved using a time-delay circuit.

## Pathway control of droplet division

Next, we applied the time-delay circuit to control the pathway of DNA droplet division (Fig. 5a). We used a ternary-mixed C·A·B-droplet, comprising three types of Y-shaped branched DNA nanostructures ($Y_C$, $Y_A$, and $Y_B$) connected with two types of linkers ($L^†_{AC}$ and $L^†_{AB}$) (Fig. 5b). $Y_A$, $Y_B$, and $L^†_{AB}$ are the same as those used in the previously described experiment; $L^†_{AC}$ was designed to crosslink $Y_C$ and $Y_A$. From the viewpoint of the reaction landscape shown in Fig. 5a, the C·A·B-droplet has two different pathways (Pathways 1 and 2) for complete division into C-, A-, and B-droplets. Pathway control was achieved by changing the inhibited division triggers (Fig. 5c). In Pathway 1, the release of $T^†_{ABi}$ is inhibited; only the C-droplet is divided from the C·A·B-droplet via cleaving $L^†_{AC}$ earlier before the complete division. In Pathway 2, the release of $T^†_{ACi}$ is inhibited; the B-droplet is divided from the C·A·B-droplet via cleaving $L^†_{AB}$ earlier.

Figures 5d and e show time-lapse images before and after adding the division triggers. To achieve Pathway 1, we added released triggers $T^†_{ACi}$ (for cleaving $L^†_{AC}$ earlier); inhibited triggers $iT^†_{ABi}$, excess inhibitors $R^†_{ABi}$, and RNase H (for cleaving $L^†_{AB}$ later). Supplementary Movie 6 shows that the order of the division of C- and B-droplets was successfully controlled, as follows. Ternary-mixed C·A·B-droplets (Fig. 5d, before addition) divided into C-droplets and binary-mixed A·B-droplets approximately 10 min after the addition (Fig. 5d (i)). After another

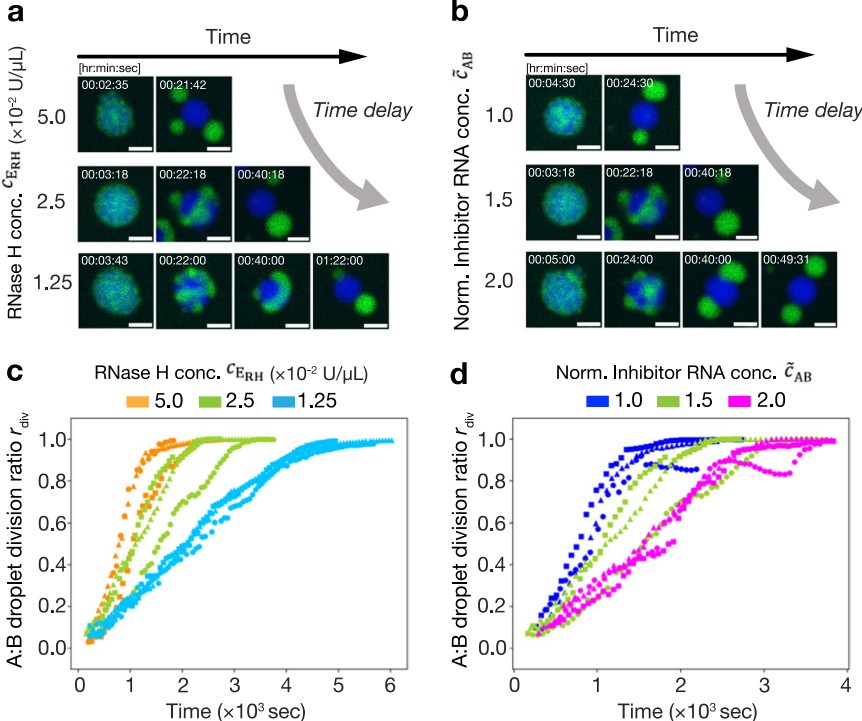

**Fig. 4 | Timing-controlled division of an A:B-droplet coupling with time-delay circuit. a, b** Time-lapse images of the division of A:B-droplets with changing the $c_{E_{RH}}$ or $u^0_{R^\dagger_{AB_i}}$ ($i = 1, 2$) (see also Supplementary Movie 1–5). Scale bars: 10 μm. **a** $c_{E_{RH}} = 1.25$, 2.5, and $5.0 \times 10^{-2}$ U/μL; $\tilde{c}_{AB} = 1.5$. **b** $c_{E_{RH}} = 2.5 \times 10^{-2}$ U/μL; $\tilde{c}_{AB} = 1.0$, 1.5, and 2.0. **c, d** Time courses of the division ratio $r_{div}$ with changing the $c_{E_{RH}}$ or $u^0_{R^\dagger_{AB_i}}$. Three repeated experiments in each condition are shown with the same color. **c** $c_{E_{RH}} = 1.25$, 2.5 and $5.0 \times 10^{-2}$ U/μL; $\tilde{c}_{AB} = 1.5$. **d** $c_{E_{RH}} = 2.5 \times 10^{-2}$ U/μL; $\tilde{c}_{AB} = 1.0$, 1.5 and 2.0. All concentrations are at the final concentration of the observed samples. Source data are provided as a Source Data file.

50 min, the A·B-droplets were divided into A- and B-droplets (Fig. 5d (ii)). This indicates that Pathway 1 was selected via the inhibition of $T^\dagger_{AB_i}$ due to the presence of $R^\dagger_{AB_i}$ (miR-6875-5p and miR-4634). Next, to achieve Pathway 2, we added released triggers $T^\dagger_{AB_i}$ (for cleaving $L^\dagger_{AB}$ earlier); inhibited triggers $iT^\dagger_{AC_i}$, excess inhibitors $R^\dagger_{AC_i}$, and RNase H (for cleaving $L^\dagger_{AC}$ later). For $R^\dagger_{AC_i}$, miRNA sequences, miR-1246 and miR-1307-3p, were used (Supplementary Table 3). Supplementary Movie 7 shows that the order of the division of B- and C-droplets was also controlled well. The C·A·B-droplets were first divided into B-droplets and C·A-droplets approximately 30 min after the addition (Fig. 5e (i)). After another 20 min, the C·A-droplets were divided into A- and C-droplets (Fig. 5e (ii)). This indicates that Pathway 2 was selected because of the presence of $R^\dagger_{AC_i}$ (miR-1246 and miR-1307-3p). Furthermore, we quantified the time courses of the division ratios of B- ($r_{div\_B}$) and C- ($r_{div\_C}$) droplets using the image processing method shown in Supplementary Note 3. The results showed that the increase of $r_{div\_B}$ was slower than $r_{div\_C}$ in Pathway 1 (Fig. 5f), while that of $r_{div\_C}$ was slower than $r_{div\_B}$ in Pathway 2 (Fig. 5g). Thus, the pathway-controlled division was achieved using time-delay circuits.

## Molecular computation: application of pathway control of droplet division

Finally, we applied the pathway control of droplet division to a molecular computing element "comparator" of RNA concentrations. Figure 6a shows the concept of the comparator using the division pathway of the C·A·B-droplet (details are explained below using Fig. 6b, c). In this comparator, Input is the initial total concentrations of miRNA sequences that are used as inhibitor RNAs in the time delay circuit. Specifically, Input 1 ($c_{AB}$) is the concentration of $R^\dagger_{AB_i}$ ($i = 1, 2$; miR-6875-5p and miR-4634) used in the time delay circuit for the delay of B-droplet division. Input 2 ($c_{AC}$) is the concentration of $R^\dagger_{AC_i}$ ($i = 1, 2$; miR-1246 and miR-1307-3p) used in the time delay circuit for the delay

of C-droplet division. Output is the selection result of the division pathway depending on the differences two Inputs ($c_{AB}$ and $c_{AC}$).

The details of the reaction scheme are shown in Fig. 6b, c. If $c_{AB} > c_{AC}$ (Fig. 6b), the $L^\dagger_{AB}$ cleavage delays longer than the $L^\dagger_{AC}$ because more $R^\dagger_{AB_i}$ causes a longer time delay of the $L^\dagger_{AB}$ cleavage; then, C-droplet is divided first, and B-droplet is divided subsequently, which means that Pathway 1 is selected. On the other hand, if $c_{AB} < c_{AC}$ (Fig. 6c), the $L^\dagger_{AC}$ cleavage delays longer; then, B-droplet is divided first, and C-droplet is divided subsequently, which means that Pathway 2 is selected. Thus, the observation of the selected pathway indicates the result of the concentration comparison between Inputs, $c_{AB}$ and $c_{AC}$.

Comparator experiments were performed using several RNA concentrations. In the experiments, we used the same DNA nanostructures as those in Fig. 5b. Here, we define $\Delta\tilde{c} = \tilde{c}_{AB} - \tilde{c}_{AC}$, where $\tilde{c}_{AB}$ and $\tilde{c}_{AC}$ are normalized initial total concentrations of inhibitor RNAs ($\tilde{c}_{AB} = u^{tot}_{R^\dagger_{AB_i}} / u^{tot}_{T^\dagger_{AB_i}}$ and $\tilde{c}_{AC} = u^{tot}_{R^\dagger_{AC_i}} / u^{tot}_{T^\dagger_{AC_i}}$ ($i = 1, 2$) are defined in the same way (see Fig. 3 caption)). We investigated five types of conditions of the initial RNA concentrations shown in Fig. 7a: $(\tilde{c}_{AB}, \tilde{c}_{AC}; \Delta\tilde{c}) = (1.25, 0; 1.25)$ (i), $(1.25, 0.75; 0.5)$ (ii), $(0.75, 1.25; -0.5)$ (iii), $(0.25, 1.25; -1.0)$ (iv), and $(0, 1.25; -1.25)$ (v). Under conditions (i)–(iii), the C-droplet divided first, whereas the B-droplet divided first under conditions (iv) and (v) (Supplementary Fig. 3 and Supplementary Movies 6–10). Figure 7a shows the time courses of the division ratios of B- ($r_{div\_B}$) and C- ($r_{div\_C}$) droplets quantified using the image processing method shown in Supplementary Note 3. These results showed that with higher $\Delta\tilde{c}$, C-droplet division was faster. Note that an increase in $\tilde{c}_t = \tilde{c}_{AB} + \tilde{c}_{AC}$ caused a delay in the overall reaction, probably because more RNA molecules induced competition in RNA degradation by RNase H in the condition of the same RNase H concentration.

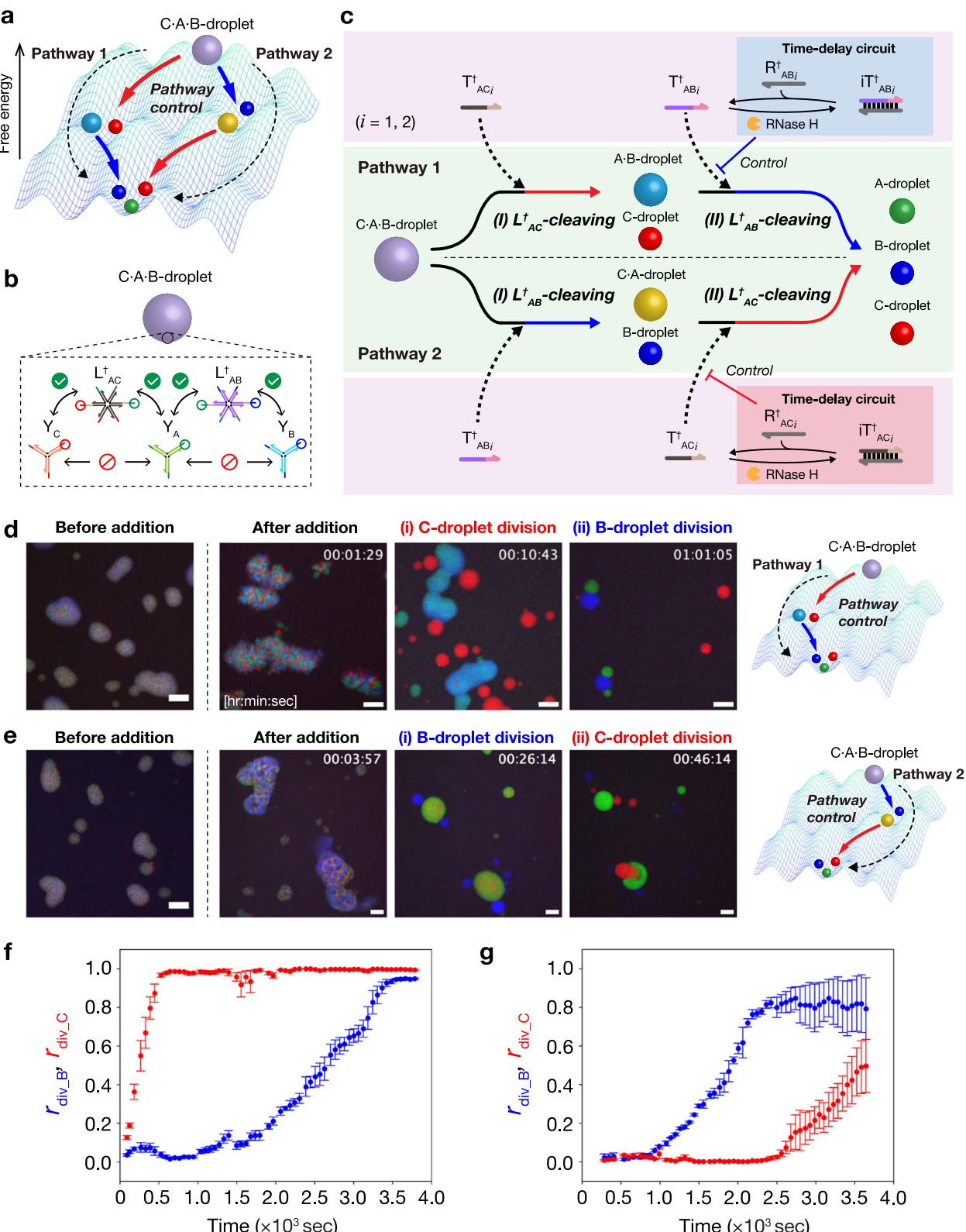

For quantitative estimation, we calculated the time difference $\Delta\tau$ between the division timings of B- and C-droplets (Fig. 7b): $\Delta\tau = \tau_{\text{div\_B}} - \tau_{\text{div\_C}}$, where $\tau_{\text{div\_B}}$ and $\tau_{\text{div\_C}}$ are defined as the times when $r_{\text{div\_B}}$ and $r_{\text{div\_C}}$ were approximately 0.5, respectively. Large errors of $r_{\text{div}}$ were observed at the later stage (Fig. 7a), which would be because the slight dissolution of droplets made background noises. Since the errors were small around $r_{\text{div}} = 0.5$, the errors did

not significantly affect the determination of $\Delta\tau$. As shown in Fig. 7b, $\Delta\tau > 0$ was observed when the RNA concentration difference $\Delta\tilde{c} = 1.25$ (i), 0.5 (ii), and $-0.5$ (iii), indicating that the division occurred through Pathway 1. Alternatively, $\Delta\tau < 0$ was observed when $\Delta\tilde{c} = -1.0$ (iv) and $-1.25$ (v), indicating that Pathway 2 was selected. These results demonstrated that the division pathway changed depending on the RNA concentration differences, confirming that the

**Fig. 5 | Control of droplet division pathway. a** A reaction landscape of the division of ternary-mixed DNA droplets. The division pathway indicates the order of droplet division. **b** Formation of ternary-mixed DNA droplet (C·A·B-droplet) containing three types of Y-shaped DNA nanostructures and two types of DNA linkers. **c** Schematic of pathway-controlled division of C·A·B-droplet. The linker-cleavage reaction rates decide the order of droplet division, thereby changing the pathway of droplet division. **d** Time-lapse images of C·A·B-droplet division in Pathway 1 before and after adding $T^{\dagger}_{ACi}$, $iT^{\dagger}_{ABi}$, $R^{\dagger}_{ABi}$, and RNase H. The detail of multistep

division process is shown in Supplementary Movie 6. $R^{\dagger}_{ABi}$ ($i$ = 1, 2): miR-6875-5p and miR-4634. Scale bars: 20 μm. **e** Time-lapse images of C·A·B-droplet division in Pathway 2 before and after adding $T^{\dagger}_{ABi}$, $iT^{\dagger}_{ACi}$, $R^{\dagger}_{ACi}$, and RNase H. The detail of multistep division process is shown in Supplementary Movie 7. $R^{\dagger}_{ACi}$ ($i$ =1,2): miR-1246 and miR-1307-3p. Scale bars: 20 μm. **f, g** Time courses of division ratio $r_{div\_B}$ (blue) and $r_{div\_C}$ (red) during C·A·B-droplet division in Pathway 1 (**f**) and Pathway 2 (**g**), respectively. Data are presented as the mean ± standard error (SE) of three field of view of microscopy observation.

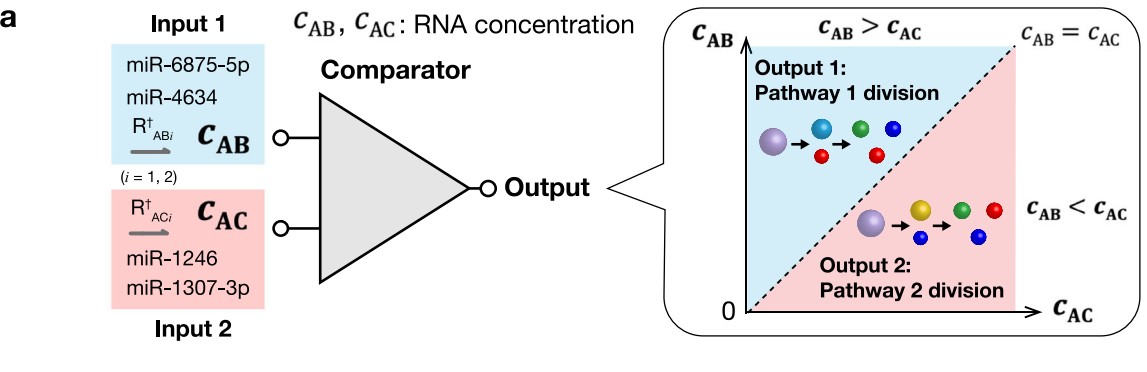

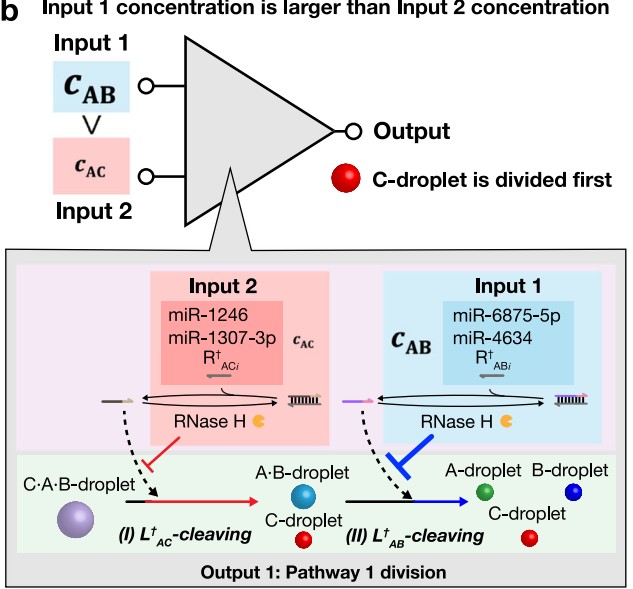

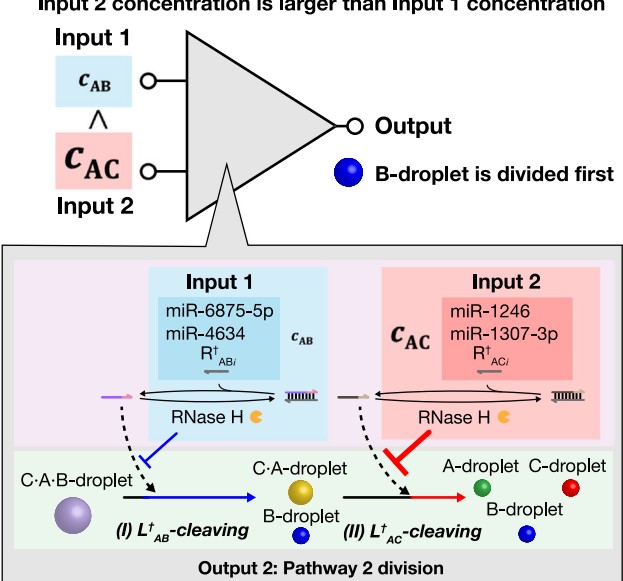

**Fig. 6 | Application of pathway control to a molecular comparator for miRNA concentrations. a** Concept of a molecular comparator of miRNA concentrations. The triangle is a symbol for a comparator element. miRNAs miR-6875-5p and miR-4634 were used for Input 1 for the comparator; miR-1246 and miR-1307-3p were used for the Input 2. The Output is the selection of the droplet division pathway, which changes depending on the difference between two initial total concentrations of miRNAs (working as inhibitor RNAs), $c_{AB}$ and $c_{AC}$. This concentration

comparison is achieved by the two time-delay circuits as shown in (**b**) and (**c**). **b** Pathway 1 is selected: if the Input 1 concentration is larger than the Input 2 concentration ($c_{AB}$>$c_{AC}$), the $L^{\dagger}_{AB}$ cleavage delays longer than the $L^{\dagger}_{AC}$ because more $R^{\dagger}_{ABi}$ causes a longer time delay of the $L^{\dagger}_{AB}$ cleavage. Thus, C-droplet is divided first, and B-droplet is divided subsequently. **c** Pathway 2 is selected: if $c_{AB}$<$c_{AC}$, the $L^{\dagger}_{AC}$ cleavage delays longer. Thus, B-droplet is divided first, and C-droplet is divided subsequently.

concentration comparator for the miRNA sequences worked as expected.

Ideally, the sign of $\Delta\tau$ is expected to switch when $\Delta\tilde{c}$ = 0 (i.e., $c_{AB}$ = $c_{AC}$). However, the results imply that the sign switches between $-1.0<\Delta\tilde{c}< -0.5$ (i.e., $c_{AB}$≠$c_{AC}$). Here, we define an offset concentration of this molecular comparator, $\sigma$, at which the sign of $\Delta\tau$ switches, where the output of the comparator switches. Ideally, $\sigma$ = 0 as shown in Fig. 6a, while our molecular comparator had a non-zero offset ($\sigma$≠0); the $\sigma$ value was estimated around $-0.75$ since the sign of $\Delta\tau$ switches between $-1.0<\Delta\tilde{c}< -0.5$ (Fig. 7c). Generally, regular electrical comparators also have a non-zero offset voltage because of non-ideal

circuit properties; similarly, our molecular comparator would have had non-ideal reaction properties. We guess that $\sigma$≠0 would be caused probably because B-droplet division took longer than that of the C-droplet for some reasons; for example, the DNA sequence difference induced the slower cleavage of $L^{\dagger}_{AB}$ than $L^{\dagger}_{AC}$, or more linker cleavage is required for B-droplet division than C-droplet division. In future studies, $\sigma$ may be tuned by sequence designing of DNAs.

To estimate the hypothesis for the mechanism of the non-zero offset, we performed numerical simulations using a reaction-diffusion model that considered differences in the cleavage rate of linker DNAs (see Supplementary Note 2). First, we changed the hybridization and

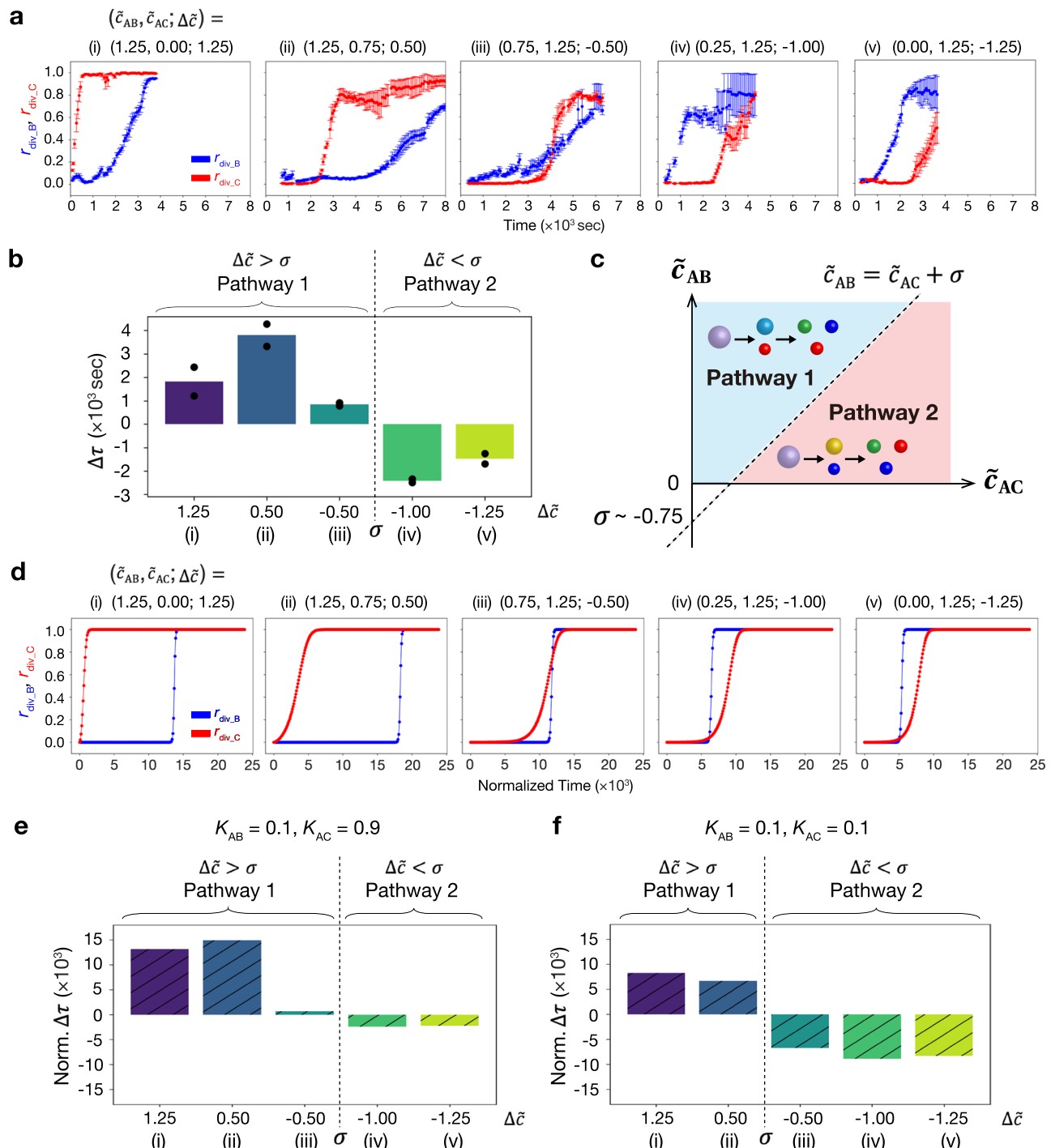

**Fig. 7 | Experimental and simulation results of molecular concentration comparator. a** Time courses of $r_{\mathrm{div\_B}}$ (blue) and $r_{\mathrm{div\_C}}$ (red) at varying the two normalized initial total concentrations of inhibitor RNAs $\tilde{c}_{\mathrm{AB}}$ and $\tilde{c}_{\mathrm{AC}}$ in the experiment. $\tilde{c}_{\mathrm{AB}} = u_{\mathrm{R^\dagger_{ABi}}}^{\mathrm{tot}} / u_{\mathrm{T^\dagger_{ABi}}}^{\mathrm{tot}}$ and $\tilde{c}_{\mathrm{AC}} = u_{\mathrm{R^\dagger_{ACi}}}^{\mathrm{tot}} / u_{\mathrm{T^\dagger_{ACi}}}^{\mathrm{tot}}$ ($i = 1, 2$), where the input initial total RNA concentrations are defined as $u_{\mathrm{R^\dagger_{AB1}}}^{\mathrm{tot}} = $ [miR-6875-5p], $u_{\mathrm{R^\dagger_{AB2}}}^{\mathrm{tot}} = $ [miR-4634], and $u_{\mathrm{R^\dagger_{AB1}}}^{\mathrm{tot}} = u_{\mathrm{R^\dagger_{AB2}}}^{\mathrm{tot}}$; $u_{\mathrm{R^\dagger_{AC1}}}^{\mathrm{tot}} = $ [miR-1246], $u_{\mathrm{R^\dagger_{AC2}}}^{\mathrm{tot}} = $ [miR-1307-3p], and $u_{\mathrm{R^\dagger_{AC1}}}^{\mathrm{tot}} = u_{\mathrm{R^\dagger_{AC2}}}^{\mathrm{tot}}$. The $\triangle\tilde{c}$ ($= \tilde{c}_{\mathrm{AB}} - \tilde{c}_{\mathrm{AC}}$) was varied at (i) 1.25, (ii) 0.50, (iii) −0.50, (iv) −1.00, and (v) −1.25. RNase H concentration was fixed at 0.25 U/μL in all experiments. The plots in conditions (i) and (v) are identical to those in Figs. 5f and 5g, respectively. Data are presented as the mean ± SE of three field of view of microscopy observation. **b** Time difference $\Delta\tau$ at each of five RNA conditions (i)-(v) in the experiment. Data are presented as the mean; more than four field of view of microscopy observation in two independent experiments. **c** Schematic of the pathway selection depending on the $\tilde{c}_{\mathrm{AB}}$, $\tilde{c}_{\mathrm{AC}}$, and offset concentration $\sigma$ in the experiment. $\sigma$ was estimated as

−0.75, which is the average of $\Delta\tilde{c}$ between conditions (iii) and (iv). **d** Time courses of $r_{\mathrm{div\_B}}$ (blue) and $r_{\mathrm{div\_C}}$ (red) at varying inhibitor RNA concentrations in the reaction-diffusion simulation. The $\Delta\tilde{c}$ was varied at (i) 1.25, (ii) 0.50, (iii) −0.50, (iv) −1.00, and (v) −1.25. The hybridization rate and the strand displacement rate between $\mathrm{T^\dagger_{ABi}}$ and $\mathrm{L^\dagger_{AB}}$ were set 10 times lower than those between $\mathrm{T^\dagger_{ACi}}$ and $\mathrm{L^\dagger_{AC}}$, respectively. Threshold parameters $K_{\mathrm{AB}}$ and $K_{\mathrm{AC}}$ were set as 0.1 and 0.9, respectively. $n = 16$. **e, f** Time difference $\Delta\tau$ at each of five RNA conditions (i)-(v) in the reaction-diffusion simulation. The hybridization rate and the strand displacement rate between $\mathrm{T^\dagger_{ABi}}$ and $\mathrm{L^\dagger_{AB}}$ were set 10 times lower than those between $\mathrm{T^\dagger_{ACi}}$ and $\mathrm{L^\dagger_{AC}}$, respectively. $K_{\mathrm{AB}} = 0.1$ and $K_{\mathrm{AC}} = 0.9$ ($\sigma \neq 0$). $n = 16$ (**e**). The hybridization rate and the strand displacement rate between $\mathrm{T^\dagger_{ABi}}$ and $\mathrm{L^\dagger_{AB}}$ were the same as those between $\mathrm{T^\dagger_{ACi}}$ and $\mathrm{L^\dagger_{AC}}$, respectively. $K_{\mathrm{AB}} = 0.1$ and $K_{\mathrm{AC}} = 0.1$ ($\sigma = 0$). $n = 16$ (**f**). Source data are provided as a Source Data file.

the strand displacement rates for $L^{\dagger}_{AB}$ cleavage. Next, we varied the threshold parameters $K_{AB}$ and $K_{AC}$ for $r_{div\_B}$ and $r_{div\_C}$ (Eqs. S.90 and S.91 in Supplementary Note 2); the larger the threshold parameters, the faster the division.

We set the hybridization rate and the strand displacement rate between $T^{\dagger}_{ABi}$ and $L^{\dagger}_{AB}$ to be 10 times lower than that between $T^{\dagger}_{ACi}$ and $L^{\dagger}_{AC}$. $K_{AB}$ and $K_{AC}$ are set to asymmetric values of 0.1, and 0.9, respectively. Figure 7d shows the time courses of $r_{div\_B}$ and $r_{div\_C}$ in the simulation results. As $\Delta\tilde{c}$ increased, the C-droplets tended to divide earlier. Additionally, as shown in Fig. 7e, the offset concentration $\sigma$ was approximately $-0.75$, indicating that the trend is consistent with the experimental result. These results suggest that the differences in the cleavage rate between $L^{\dagger}_{AB}$ and $L^{\dagger}_{AC}$ and the required amount of linker cleavage for B-droplet and C-droplet divisions resulted in $\sigma \neq 0$. Furthermore, numerical simulations were performed using different parameter values (Fig. 7f and Supplementary Figs. 4–7), producing different offset concentrations. These results suggest that changing DNA sequences could potentially control the offset concentration $\sigma$. Note that, when more $\tilde{c}_t$, the simulation results reproduce the delay in the overall reaction as observed in experiments due to the competition in the RNase H reaction.

## Discussion
We demonstrated the timing-controlled division dynamics of DNA droplets using a time-delay circuit. We developed the reaction-diffusion model and numerically investigated the strategy to control the division timing by controlling the cleavage rate of $L^{\dagger}_{AB}$. Using this strategy, we experimentally demonstrated timing control of the division of an A:B-droplet by tuning the time-delay circuit parameters. Although the current simulation model focused on the spatiotemporal distribution of linker DNAs to estimate the time-delay circuit behavior, the model would be extended to a model explicitly considering the phase separation process by adding the Cahn-Hilliard term[58]. In addition, although our model suggested one of the possibilities of the cause of the offset concentration σ, this model is not perfect as discussed above; thus, further study on numerical modeling with experimental studies would be required.

Using the time-delay circuit, we realized the pathway control of the C·A·B-droplet division by changing the order of two types of linker DNA cleavage. Finally, the pathway control of the C·A·B-droplet division was employed for molecular computation. We achieved not only the detection of the presence/absence of miRNA sequences but also the comparison of the concentrations of miRNA sequences, which may be applied to a diagnosis based on the expressed miRNA concentrations. Based on these results, we revealed that the RNase H-based strategy could be applied to the control of the phase separation dynamics as well as the other DNA nanotechnologies in a bulk solution.

The RNA concentration comparator had non-zero offset ($\sigma \neq 0$) (Fig. 7b), and the simulation results suggested that σ changed depending on hybridization rates or strand displacement rates of linker DNAs (Fig. 7e, f and Supplementary Figs. 4–7). Because the hybridization and strand displacement rates of DNAs depend on their sequence and length[55], these results suggest that the non-zero offset was probably due to the sequences of the linker DNA nanostructure. Previously, Nguyen et al. [59] and Sato et al. [57] have shown that differences in the sequences of DNA nanostructures changed the kinetic and thermodynamic properties of DNA droplets. To further control the DNA droplet dynamics, the influence of DNA sequences on the kinetic properties of DNA nanostructures must be clarified.

The present study demonstrated that chemical reactions could control DNA droplet dynamics such as droplet division. However, since the coupled chemical reactions were only a transient non-equilibrium relaxation process, far-from-equilibrium chemical reactions with

sustained chemical energy supplies are required to achieve truly active systems. Moreover, in future, the control of chemical reactions via the physical dynamics of DNA droplets and the reversible control of DNA droplet dynamics should be explored. Such bidirectional control over more complex dynamics can help build artificial cells with more living cell-like functions, such as biochemical reactions controlled by the condensates of transcriptional factors and cell/organelle behaviors controlled by transcripts[6,60]. Moreover, enzymatic reactions regulated by synthetic protein-based coacervates[61] can be combined with our DNA-based droplet system. We believe that this technology provides a strategy to create artificial cells and molecular robots with more sophisticated functions, such as timing-controlled self-replication, drug delivery, and diagnosis, with more accuracy and quantitative specifications.

## Methods
### Sequence design and oligo-nucleotides preparation
DNA and RNA sequences were designed using the Nucleic Acid Package (NUPACK)[62]. DNA sequences listed in Tables S1–S3 were purchased from Eurofins Genomics (Tokyo, Japan). The fluorescently labeled DNA was purified using high-performance liquid chromatography (HPLC), while the others were purified using an oligonucleotide purification cartridge (OPC). RNA sequences listed in Table S3 were purchased from Sangon Biotech (Shanghai, China) and purified using HPLC. The purchased oligonucleotide powders were diluted to 100 or 200 µM with ultra-pure water (Direct-QUV, Millipore, ZRQSVP030) and stored at −20 °C.

### Preparation of mixed DNA droplets
We prepared three DNA droplets (A·B-droplet, A:B-droplet, C·A·B-droplet). In Figs. 2c, g, a sample solution for the A·B-droplet contained 5 µM $Y_A$, 5 µM $Y_B$, and 1.65 µM $L_{AB}$ in a reaction buffer (20 mM Tris-HCl [pH 8.0], 350 mM NaCl) was heated at 85 °C for 5 min and then cooled down from 85 °C to 25 °C at a rate of −1 °C/min to anneal the contained DNAs using a thermal cycler (Mastercycler® nexus X2, Eppendorf, Germany). In Figs. 4a, b, a sample solution for the A:B-droplet contained 5 µM $Y_A$, 5 µM $Y_B$, 1.485 µM $L_{AB}$, and 0.165 µM $L^{\dagger}_{AB}$ in the reaction buffer was heated and cooled down in the same manner. In Fig. 5d, e, a sample solution for the C·A·B-droplet contained 1.0 µM $Y_A$, 1.0 µM $Y_B$, 1.0 µM $Y_C$, 2.0 µM $L^{\dagger}_{AB}$, and 2.0 µM $L^{\dagger}_{AC}$ in the reaction buffer was heated and cooled down in the same manner. After annealing, the sample of the A·B-droplet and A:B-droplet were diluted twofold with the reaction buffer. The C·A·B-droplet was not diluted. The concentrations of each strand in the mixed DNA droplet after dilution and addition of the division trigger mixture are shown in Supplementary Tables 4–6. Tris-HCl (pH 8.0) (cat. #15568025) was purchased from Invitrogen (Carlsbad, CA), and NaCl (cat. #191-01665) was purchased from Wako (Japan), respectively.

### Microscopy observation
To observe the autonomous division of the A·B-droplet samples and the timing-controlled division of the A:B-droplet samples, we used a confocal laser scanning microscopy (CLSM) (FV-1000, Olympus, Tokyo, Japan) and a stage heater (10021-PE120 system, Linkam, Fukuoka, Japan). To observe the pathway-controlled division of the C·A·B-droplet, we used fluorescent microscopy (IX-71, Olympus, Tokyo, Japan) equipped with a spinning-disk confocal system (CSU-X1, Yokogawa, Tokyo, Japan), an EM CCD camera (iXon X3, Andor), and the stage heater. Samples containing 6-FAM, Alexa 405, and Cy3 were visualized at excitation wavelengths of 473, 405, and 561 nm, respectively. Observation chambers were prepared for CLSM observation. Glass slides (dimensions: 30 × 40 mm, thickness: 0.17 mm, Matsunami, Kishiwada, Japan) were soaked in 5% bovine serum albumin (BSA) (cat. #019-15123, Wako, Japan) solution with 20 mM Tris-HCl (pH 8.0) for 30 min. After BSA coating, the glasses were washed with ultrapure

water and dried. The 1-mm-thickness silicon sheet (cat. #107-0040202, Kokugo, Japan) with 5 mm-diameter holes was placed on the BSA-coated glass.

## Autonomous division experiments of A·B-droplets

The A·B-droplet sample solution containing 5 μM $Y_A$, 5 μM $Y_B$, and 1.65 μM $L_{AB}$ in a reaction buffer (20 mM Tris-HCl [pH 8.0], 350 mM NaCl) was heated 85 °C for 5 min and then cooled down from 85 °C to 25 °C at a rate of −1 °C/min to anneal the contained DNAs using a thermal cycler. The trigger mixture comprised 2.5 μM $T_{AB1}$ and 2.5 μM $T_{AB2}$ in the reaction buffer. A·B-droplet sample solution (3 μL) was placed in the 5 mm hole of the observation chamber. The sample solutions were covered with mineral oil to prevent evaporation. The chamber was incubated on a stage heater at 60 °C for 30 min to increase the fluidity of the DNA droplets. After incubation, we added 3 μL of the trigger mixture to the sample solution in the chamber and observed it at 60 °C.

## Timing-controlled division experiments of A:B-droplets by adding a division trigger solution

The A:B-droplet sample solution containing 5 μM $Y_A$, 5 μM $Y_B$, 1.485 μM $L_{AB}$, and 0.165 μM $L^{\dagger}_{AB}$ in the reaction buffer was heated and cooled down in the same manner. The trigger mixture was prepared by mixing $T_{AB}$ solution, $iT^{\dagger}_{AB}$ solution, RNase H solution, and $MgCl_2$ solutions. The $T_{AB}$ solution comprised 25 μM $T_{AB1}$ and 25 μM $T_{AB2}$ in the reaction buffer. To prepare the $iT^{\dagger}_{AB}$ solution, we mixed ssDNA triggers ($T^{\dagger}_{AB1}$, $T^{\dagger}_{AB2}$), and inhibitor RNAs ($R^{\dagger}_{AB1}$ and $R^{\dagger}_{AB2}$) in the reaction buffer. Here, the concentration of $T^{\dagger}_{AB1}$ and $T^{\dagger}_{AB2}$ ($u^{tot}_{T^{\dagger}_{ABi}}$, ($i=1, 2$)) was fixed at 1.0 μM, while that of $R^{\dagger}_{AB1}$ and $R^{\dagger}_{AB2}$ ($u^{tot}_{R^{\dagger}_{ABi}}$, ($i=1, 2$)) was changed to 1.0, 1.5, and 2.0 μM at a final concentration. Normalized initial total RNA concentration $\widetilde{c}_{AB} = u^{tot}_{R^{\dagger}_{ABi}}/u^{tot}_{T^{\dagger}_{ABi}}$ is defined. The $iT^{\dagger}_{AB}$ solution was heated at 85 °C for 5 min and then cooled down from 85 °C to 25 °C at a rate of −1 °C/min to form RNA-DNA hybrids. The RNase H solution contained thermostable ribonuclease H (RNase H) (M0523S, New England Biolabs) in the reaction buffer. The concentrations of RNase H used were 0.125, 0.25, and 0.5 U/μL. The $MgCl_2$ solution comprised 15 mM $MgCl_2$ (20908-65, Nacalai Tesque, Japan) in the reaction buffer. To prepare the trigger mixture, the $T_{AB}$ solution, $iT^{\dagger}_{AB}$ solution, RNase H solution, and $MgCl_2$ solution were mixed with the reaction buffer at the concentrations shown in Supplementary Table 7. The normalized initial total concentration of inhibitor RNA is defined as $\widetilde{c}_{AB} = u^{tot}_{R^{\dagger}_{ABi}}/u^{tot}_{T^{\dagger}_{ABi}}$ ($i=1, 2$), where $u^{tot}_{R^{\dagger}_{ABi}}$ is the concentration of $R^{\dagger}_{ABi}$, and $u^{tot}_{T^{\dagger}_{ABi}}$ is the concentration of $T^{\dagger}_{ABi}$.

A:B-droplet sample solution (3 μL) was placed in a 5 mm hole of the observation chamber. The sample solutions were covered with mineral oil to prevent evaporation. The chamber was incubated on a stage heater at 60 °C for 30 min to increase the fluidity of the DNA droplets. After incubation, we added 3 μL of the trigger mixture to the sample solution in the chamber and observed it at 60 °C. As shown in Supplementary Table 7, when the final concentration of $c_{E_{RH}}$ was fixed at $2.5 \times 10^{-2}$ U/μL, $\widetilde{c}_{AB}$ was varied as 1, 1.5, and 2. When $\widetilde{c}_{AB}$ was fixed at 1.5, $c_{E_{RH}}$ was varied as $1.25 \times 10^{-2}$, $2.5 \times 10^{-2}$, and $5.0 \times 10^{-2}$ U/μL. To calculate the division ratio, $r_{div}$, we binarized the fluorescent images and analyzed them using Fiji[63].

## Pathway-controlled division experiments of C·A·B-droplets by adding a division trigger solution

To demonstrate the pathway-controlled division of C·A·B-droplets in Figs. 5d, e, we added the trigger mixture to the C·A·B-droplet sample solution. Trigger mixtures 1 and 2 were prepared for the division pathways 1 and 2, respectively. The trigger mixture 1 was prepared by mixing $T^{\dagger}_{AC}$, $iT^{\dagger}_{AB}$, RNase H, and $MgCl_2$ solutions. The $T^{\dagger}_{AC}$ solution comprised 30 μM $T^{\dagger}_{AC1}$ and 30 μM $T^{\dagger}_{AC2}$ in the reaction buffer. To

prepare the $iT^{\dagger}_{AB}$ solution, we mixed ssDNA triggers $T^{\dagger}_{AB1}$ and $T^{\dagger}_{AB2}$ (6 μM each) and inhibitor RNAs $R^{\dagger}_{AB1}$ and $R^{\dagger}_{AB2}$ (7.5 μM each) in the reaction buffer. The $iT^{\dagger}_{AB}$ solution was heated 85 °C for 5 min and then cooled down from 85 °C to 25 °C at a rate of −1 °C/min to form RNA-DNA hybrids. The RNase H solution contained 2.0 U/μL RNase H in the reaction buffer. The $MgCl_2$ solution comprised 30 mM $MgCl_2$ in the reaction buffer. To prepare the trigger mixture 1, the $T^{\dagger}_{AC}$ solution, the $iT^{\dagger}_{AB}$ solution, the RNase H solution, and the $MgCl_2$ solution were mixed with the same buffer at the concentrations shown in Supplementary Table 8.

The trigger mixture 2 was prepared by mixing $T^{\dagger}_{AB}$, $iT^{\dagger}_{AC}$, RNase H, and $MgCl_2$ solutions. The $T^{\dagger}_{AB}$ solution was composed of 30 μM $T^{\dagger}_{AB1}$ and 30 μM $T^{\dagger}_{AB2}$ in the reaction buffer. To prepare the $iT^{\dagger}_{AC}$ solution, we mixed ssDNA triggers $T^{\dagger}_{AC1}$ and $T^{\dagger}_{AC2}$ (6 μM each) and inhibitor RNAs $R^{\dagger}_{AC1}$ and $R^{\dagger}_{AC2}$ (7.5 μM each) in the reaction buffer. The $iT^{\dagger}_{AC}$ solution was heated 85 °C for 5 min and then cooled down from 85 °C to 25 °C at a rate of −1 °C/min to form RNA-DNA hybrids. The RNase H and $MgCl_2$ solutions were the same as those used to prepare trigger mixture 1. To prepare the trigger mixture 2, the $T^{\dagger}_{AB}$ solution, the $iT^{\dagger}_{AC}$ solution, the RNase H solution, and the $MgCl_2$ solution were mixed with the same buffer at the concentrations shown in Supplementary Table 9.

For the RNA concentration comparator experiments, we varied the RNA concentration in the trigger mixture. The trigger mixture was prepared by mixing the $iT^{\dagger}_{AB}$ solution, $iT^{\dagger}_{AC}$ solution, an RNase H solution, and an $MgCl_2$ solution. The inhibitor RNA concentration of $iT^{\dagger}_{AB}$ solution and $iT^{\dagger}_{AC}$ solution were changed based on each experimental condition. The concentration of the trigger mixture after mixing it with the C·A·B-droplet sample at each experimental condition is shown in Supplementary Table 10. Normalized initial total concentration of inhibitor RNA $\widetilde{c}_{AB}$ is defined as $\widetilde{c}_{AB} = u^{tot}_{R^{\dagger}_{ABi}}/u^{tot}_{T^{\dagger}_{ABi}}$ ($i=1,2$), where $u^{tot}_{R^{\dagger}_{ABi}}$ is the concentration of $R^{\dagger}_{ABi}$, and $u^{tot}_{T^{\dagger}_{ABi}}$ is the concentration of $T^{\dagger}_{ABi}$. Normalized initial total concentration of inhibitor RNA $\widetilde{c}_{AC}$ is defined as $\widetilde{c}_{AC} = u^{tot}_{R^{\dagger}_{ACi}}/u^{tot}_{T^{\dagger}_{ACi}}$ ($i=1,2$), where $u^{tot}_{R^{\dagger}_{ACi}}$ is the concentration of $R^{\dagger}_{ACi}$, and $u^{tot}_{T^{\dagger}_{ACi}}$ is the concentration of $T^{\dagger}_{ACi}$.

C·A·B-droplet sample solution (2.4 μL) was placed in the 5 mm hole of the observation chamber. The sample solutions were covered with mineral oil to prevent evaporation. The chamber was incubated on a stage heater at 60 °C for 30 min and 63 °C for 15 min to increase the fluidity of the DNA droplets. After incubation, we added 3.6 μL of the trigger mixture to the sample solution in the chamber and observed it at 63 °C.

## Statistics and reproducibility
Data are presented as means ± SE with the number of replicates indicated. No statistical methods were used to determine sample size.

## Reporting summary
Further information on research design is available in the Nature Portfolio Reporting Summary linked to this article.

## Data availability
The dataset of the main figures generated in this study is provided in the Supplementary Information, Supplementary Videos and Source Data files. Source data are provided with this paper.

## Code availability
The source codes for numerical simulations are provided through GitHub (https://github.com/takinouelab/MaruyamaTakinoue2024)[64].

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

## Acknowledgements

We thank Mr. Ryohei Furuichi, Dr. Hirotake Udono, Mr. Nozomi Tsumura, and Dr. Marcos K. Masukawa for fruitful discussions. This work was supported by MEXT/JSPS KAKENHI (Nos. JP20H05701, JP20H00619, JP20H05935, and JP24H00070) to M.T., Human Frontier Science Program (HFSP; RGP0016/2022-102) to M.T., and JSPS Research Fellowships for Young Scientists (DC1) (Nos. JP22KJ1346) to T.M.

## Author contributions

M.T. provided the original concept. T.M. and M.T. performed all experiments. T.M., J.G., and M.T. design experiments. T.M. and M.T. wrote and revised the manuscript.

## Competing interests

The authors declare no competing interests.
