## [Peer Review File · Nature Communications]

REVIEWER COMMENTS

Reviewer #1 (Remarks to the Author):

I have read with interest the manuscript by Maruyama, Gong and Takinoue on engineering division-like behaviors in synthetic DNA droplets. This manuscript represents a valuable contribution to a growing field, and nicely exemplifies the potential of these “smart materials” to perform complex functionalities in response to external stimuli. This work builds on previous ground-breaking results from the Takinoue group on engineering division in DNA droplets and performing molecular computation with these systems, but introduces important elements of innovation, particularly the possibility of controlling the timing of division events and their sequence (in 3-component droplets). I think this work would be well received by the community interested in DNA nanotechnology and biomimetic systems. For these reasons I am happy to recommend the manuscript for publication in Nature Communications after the authors have addressed the following minor concerns.

1) In Fig. 2c and Fig. 4 a and b the mixed condensates (prior to adding the division triggers) do not appear to be fully uniform, i.e. there are areas richer in the blue and green components. Could the authors comment? Would the droplets become more uniform at higher concentrations of the linker construct? Is this marginally mixed state required to trigger division?

2) I am not sure I generally agree with the division free energy landscape in Fig. 1f. In general, the division appears to occur through a process similar to spinodal decomposition. If this is the case, once the linkers are split by the triggers, there should not be a free energy barrier separating the unstable mixed state and the stable divided state. I would remake this diagram in a way that it shows the landscape prior to adding the triggers, with a single minimum corresponding to the mixed state, and after adding the trigger, with a single minimum corresponding to the divided state.

3) I am not sure about the utility of having some of the differential equations underpinning the reaction-diffusion model in the main text, with the complete system being in SI. Perhaps having a simpler explanation in the main text (without equations) would be sufficient and possibly clearer.

4) I believe that kh_{AB} should be a strand displacement rate, rather than a hybridization rate as currently stated.

5) The patterns in Fig. 3d and e are interesting, and I think deserve further discussion. It seems that the non-delayed linkers in Fig. 3d are eliminated from the outside of the droplet towards the inside,

with a propagating front. This probably occurs because the rate of “reaction” (strand displacement disassembling the linkers) is faster than the rate of diffusion through the droplet, given the high concentration of trigger strands available. Instead, the delayed linkers in Fig. 3e appear to be eliminated uniformly throughout the droplet. This is likely because the rate of linker disassembly is in this case slower compared to the rate of diffusion, given that small amounts of trigger strands are available at any given time due to the action of the RNA blockers. Could the authors provide discussion along these lines if they agree? If not, what could be the cause of the observed difference in behaviour?

6) In Fig. 3 f and g: I find it a bit strange that the authors decided to plot the sum of the concentration of the two types of linkers. Why not show their time evolution individually rather than showing the sum? This would be more insightful.

7) In Fig. 4 c and d it is interesting that the division curves are not as sharp as one would expect from simulations. Could the authors comment on this? Could this be because of size polydispersity in the droplets? I.e. do the authors observe a correlation between droplet size and onset time of division?

8) Data in Fig. 6c become quite noisy at later times, with some sudden jumps and large error bars. Could the authors comment on the origin of these? Do they expect them to produce large errors in the determination of $\Delta\tau$?

Reviewer #2 (Remarks to the Author):

In this work, Maruyama et al. present a method for temporally controlling the division of DNA-based liquid-liquid phase separation droplets (DNA droplets). DNA droplets are fascinating because of their capability to mimic natural cellular compartments within synthetic environments. The temporal control of their division is particularly interesting as it has the potential to spatially and temporally control biochemical reactions within artificial cells.

Here, the authors achieve temporal control of DNA droplet division using a time-delay circuit, which is based on the degradation of RNA within a RNA-DNA heteroduplexes by RNase H. First, they demonstrate their previously established droplet division method, which employs trigger DNA strands that separate linker structures within the DNA droplets via toehold-mediated strand displacement reactions. Next, they present the principle of their new method, which introduces a time-delay, by using RNA strands that bind to the trigger strands effectively inhibiting the strand

displacement reaction. Upon RNA cleavage by RNase H, the trigger strands are released, enabling the separation of the linker structures. Before testing their new method, they perform theoretical simulations based on numerically solving a reaction-diffusion model. The simulations qualitatively predict the effect of varying the RNase H, as well as inhibitor RNA concentration, on the division rate of the linker structures. Afterwards, they verify their simulation results by performing experiments of the temporally controlled DNA droplet division using confocal laser scanning microscopy. Furthermore, they show that their method can be upscaled to cleave different linker structures, enabling a droplet division into distinct compartments along different pathways. Finally, they demonstrate an application of their pathway control to realize a molecular comparator for miRNA concentrations.

Overall, I enjoyed reading the manuscript, as it is well crafted and easy to follow. The experimental data is well complemented with theoretical considerations. Below are my specific comments regarding the manuscript:

Major comments:

1. In the first results section, the authors should state more clearly, that their DNA droplet design as well as division triggering follows their previous work.
2. The definition of the droplet division ratio r_{div} (Supplementary Note S3) seems to not match to what is shown in Figures 4c,d and Figure 6c. Given the current definition, for perfectly mixed droplets, the number of pixels having both fluorophores (N_{AB}) should match the number of pixels having the blue fluorophore (N_B), thus the ratio is 1. For perfectly separated droplets, the number of pixels having both fluorophores should be 0, resulting in a ratio of 0. What I think is shown, is $r_{div} = 1 - N_{AB}/N_B$. To resolve this discrepancy, the authors should revise their definition of the droplet division ratio.
3. There is a discrepancy within the value of the threshold concentration K . In the main text on page 10, a K value of 0.95 is stated to be shown in Figure 3, whereas in supplemental Figure S2, the K value was changed between 0.01, 0.05 and 0.1. It looks like $n=16$ and $K=0.05$ of Figure S2b are shown in Figure 3 h,i. I think this might be due to the threshold concentration once relating to the total linker concentration and once to the linker concentration that can be inhibited? The authors should correct this discrepancy.
4. Comparing Figure 3 h,l and Figure 4 c,d, the experimental observations do not align well with the predicted sigmoidal shape of the theoretical simulations. The authors should discuss this discrepancy. Furthermore, why was a ratio of 90% to 10% chosen for linkers that are not inhibited to linkers that can be inhibited? Would a higher amount of linkers that can be inhibited lead to a more sigmoidal shape as predicted by the simulations?
5. The authors should quantify their observation for the control of the droplet division pathway (Figure 5c,d) similar to Figure 6c.
6. Figure 6c shows much better division delays compared to Figure 4. The authors should provide supplementary videos showing some of these reactions.

Minor comments:

1. The supplementary videos provide valuable visual insight into the temporal control of the DNA droplets. Unfortunately, some of the traces exhibit noticeable drift or abrupt jumps along the time trajectory, making it hard to follow single droplets. Since the videos are already processed within Fiji, I suggest considering the implementation of a simple drift correction using a plugin such as StackRegJ, Fast4DReg, or the manual drift correction plugin. This could enhance the overall quality of the presented data.
2. Overall, the experiments and conclusions are well and logically described and clearly understandable. However, the manuscript would benefit from improving the grammar of the text.
3. Page 9 line 5: the period should be behind the “”, to read: ”X”.
4. The term “decreasing rate” is used to describe the decrease in concentration of the Linkers. It becomes confusing when used together with decreasing and increasing concentrations. I suggest to use more specific terms such as “LAB cleavage rate” or even better “LAB cleavage kinetics” since the simulations provide a kinetics rather than a rate.
5. I also suggest to avoid the usage of formula symbols when discussing concentrations and other parameters in the text (e.g. decreasing c_{Erh} or increasing u_{RABi}) wherever this is possible. This makes it easier to follow the text.

Reviewer #3 (Remarks to the Author):

Maruyama et al. present a method to achieve division dynamics in artificial DNA-based droplets by using multiple time-delayed triggers controlled by out-of-equilibrium chemical reactions. First, they generate binary mixed DNA droplets by combining two Y-shaped DNA nanostructures, YA and YB, connected by 6-branched DNA linkers (A:B droplets). Division is achieved by cleavage of the linkers through hybridization with division-triggering DNAs. The main goal of the project is to develop a time-delay circuit with a reaction that prevents division by hybridization with excess single-stranded inhibitor RNAs. This leads to a temporal delay of linker cleavage and consequently the division of DNA droplets by RNA degradation with ribonuclease H (RNase H). Finally, they explored an alternative method for generating C-A-B droplets using two different types of DNA cleavage linkers, which enabled the construction of a molecular comparator for miRNA concentrations.

The proposed approach is clever in itself and offers an interesting opportunity to specifically divide DNA condensates. The manuscript and the SI are generally well organized; the claims about the timed division dynamics are supported by the data, and by what one would predict based on previous work by the authors and others.

My main comments are below.

1) While the use of RNase H for the temporal control of DNA-based LLPS droplets is innovative, the overall novelty of the concept is perhaps limited. First, the authors already used enzymes cleaving 6 arm nanostars to demonstrate droplet division (Sci Adv 2020, 6 (23), eaba3471). The use of DNA triggers is its natural extension, given recent work showing how strand displacement/invasion can be used to modify the properties of DNA nanostar-based condensates (Science Advances 8.41 (2022): eabj1771 and Nature Communications 15.1 (2024): 1915). Previous work has also reported that the timing of DNA-based reactions and assemblies can be tuned through the use of RNase H (JACS 145.38 (2023): 20968-20974, JACS 143.48 (2021): 20296-20301). In addition, the use of pathway control for molecular computation and comparator of miRNA concentrations may not be particularly groundbreaking, as the same group has shown in 2022 (Adv. Funct. Mater. 2022, 2202322) that detection of miRNA is possible through the development of DNA-responsive droplets.

2) The model is a nice addition, although it is quite complex and therefore difficult to follow - it is well formulated as far as I can tell. I was initially a bit confused about the use of a purely diffusive term in the PDE, instead of a Cahn Hilliard term that is appropriate for phase separation, but I understand that since the total concentration of monomers is actually not changing, and there is no phase transition, this simplified approach is ok. A question I have is the choice of a very high Hill coefficient ($n=16$) for modeling division ratio as a function of the trigger, which I think is why the authors get very steep division curves in Fig. 3h and i.

In these plots, changing RNase H level or inhibitor concentration creates a finite delay without altering the slope of the curve. In contrast, experiments in Fig. 4 c and d show that if RNase H and inhibitor level are changed, there is no clear delay while the slope of the division rate changes. Even in the fastest case (orange curves), the slope of the simulations is much larger when compared to the data. I think the Hill coefficient that was chosen is too large; if I understand correctly, one needs two copies of trigger to split one nanostar, so I would expect the Hill coefficient to be 2. Why did the authors choose $n=16$? Perhaps there are aspects of how data were normalized that I do not understand, but the most important output of the model right now behaves quite differently from the data.

3) I find the overall narrative of the paper to be somewhat controversial. I think that when DNA triggers are added, there is no active, autonomous process happening as the authors claim. The overall thermodynamic landscape of the system is changed by the new DNA, and the ensemble of strands just moves to a new equilibrium. I don't understand how this is active, or how any fuel is used to maintain the system out of equilibrium. The old equilibrium no longer exists given how the system is designed. My concept of a non-equilibrium system is one where a fuel molecule maintains the system to an otherwise energetically unfavorable equilibrium state; when fuel is removed from the system, it will relax to its original, energetically stable equilibrium. In this sense,

triggers are not a fuel molecule, they are just new components that change the equilibrium to a new resting state that didn't exist before. Conceptually, I think that if the division trigger strands were RNA, degraded by RNase H (which should be present from the start), then one could say that as long as RNA fuel is present, then the system is maintained in an out-of-equilibrium state (split A/B droplets). As soon as RNase H is done degrading the RNA fuel, then the system would revert to the original equilibrium (mixed A/B droplets).

4) Could the division reactions be reversed and how? Can the original conformation of the mixed droplets be regained starting from split droplets?

5) I do not understand how the comparator circuit works, in the main paper there is not enough detail. Can the authors please clarify with a schematic how the "triangle" in Fig. 6a works? What part of the nanostar/linker strands had to be redesigned, if any, to become responsive to the miRNA sequences?

Reviewer #3 (Remarks on code availability):

The code is provided in python, which I normally don't use. I have not tried to run it myself.

It could be commented a bit more. Also the authors should make sure that all comments are in both Japanese and English. Right now they are mostly in Japanese.

Reviewer #4 (Remarks to the Author):

Reviewer 1

I have read with interest the manuscript by Maruyama, Gong and Takinoue on engineering division-like behaviors in synthetic DNA droplets. This manuscript represents a valuable contribution to a growing field, and nicely exemplifies the potential of these “smart materials” to perform complex functionalities in response to external stimuli. This work builds on previous ground-breaking results from the Takinoue group on engineering division in DNA droplets and performing molecular computation with these systems, but introduces important elements of innovation, particularly the possibility of controlling the timing of division events and their sequence (in 3-component droplets). I think this work would be well received by the community interested in DNA nanotechnology and biomimetic systems. For these reasons I am happy to recommend the manuscript for publication in Nature Communications after the authors have addressed the following minor concerns.

We appreciate the reviewer’s overall positive evaluation. We have revised our manuscript based on the reviewer’s fruitful comments and questions as follows.

1) In Fig. 2c and Fig. 4 a and b the mixed condensates (prior to adding the division triggers) do not appear to be fully uniform, i.e. there are areas richer in the blue and green components. Could the authors comment? Would the droplets become more uniform at higher concentrations of the linker construct? Is this marginally mixed state required to trigger division?	According to the previous study (Adv. Funct. Mater. 2022, 2202322), increasing the amount of DNA linkers may lead to a fully uniform state. However, empirically, the marginally mixed state is observed in a mixed DNA droplet containing many types of DNA nanostructures even when increasing the amount or stability of linkers. Additionally, the marginally mixed state is not required for the division since DNA droplets in a fully uniform state were also able to divide, as observed in the previous study (Adv. Funct. Mater. 2022, 2202322). We have reflected the reviewer's comments by adding the following sentence: (p.6) “The result agreed with the previous study⁴⁷; although a slightly inhomogeneous area richer in Y_A or Y_B component was observed in the A·B-droplet, the inhomogeneity was not necessary for a droplet division.”
2) I am not sure I generally agree with the division free energy landscape in Fig. 1f. In general, the division appears to occur through a process similar to spinodal decomposition. If this is the case, once the linkers are split by the triggers, there should not be a free energy barrier separating the unstable mixed state and the stable divided state. I would remake this diagram in a way that it shows the landscape prior to adding the triggers, with a single minimum corresponding to the mixed state, and after adding the trigger, with a single minimum corresponding to the divided state.	We thank the reviewer for pointing out the very important point. We agree with the reviewer's comments on spinodal decomposition. Our previous free energy landscape shows only the two states of the DNA linker before and after its cleavage. The A·B-droplet division occurs after the linker cleavage through the spinodal-decomposition manner as the reviewer indicated. That's why our diagram confused the reviewer. To avoid such confusion, we remade the free energy landscape including both the linker cleavage and spinodal decomposition for phase separation as below. The new free energy landscape has three states: (i) A·B-droplet and ssDNA fuels outside the droplet, where the DNA linker is not cleaved yet. (ii) A·B-droplet with ssDNA fuels inside, where DNA linker is cleaved but the DNA droplet is not divided yet; (iii) A- and B-droplets are separated after the spinodal decomposition.

As the reviewer indicated, the process between (ii) and (iii) has no barrier (i.e., single minimum) during the spinodal-decomposition-based phase separation.

We changed the Figure 1 caption: “Description of the A·B-droplet division dynamics based on reaction landscapes. The ssDNA division triggers work as “fuel” molecules, which change the reaction landscape from a single-minimum shape to a double-minimum shape with three transition steps. (i) A·B-droplet with ssDNA fuels outside, where the DNA linker is not cleaved yet. (ii) A·B-droplet with ssDNA fuels inside, where DNA linker is cleaved but the DNA droplet is not divided yet; (iii) A- and B-droplets are divided through the spinodal decomposition. ΔG_{CIV} and ΔG_{PS} are Gibbs free energy changes for the linker cleavage reaction and the phase separation, respectively.”

3) I am not sure about the utility of having some of the differential equations underpinning the reaction-diffusion model in the main text, with the complete system being in SI. Perhaps having a simpler explanation in the main text (without equations) would be sufficient and possibly clearer.

We thank the reviewer for the suggestion.

We intended to explain that the time-delay circuit has two important factors to control the delay: the total RNase H concentration c_{ERH} , and the initial concentrations of excess inhibitor RNAs $u_{\text{R}^\dagger_{\text{AB}i}}^0$. To follow the reviewer's suggestion, we simplified the equations by deleting the differential equation for linkers and simplified the explanation as follows:

(p.9)

“First, we numerically investigated the dependence of the cleaving rate of the DNA linker L^\dagger_{AB} on the concentrations of RNase H and the inhibitor RNAs ($R^\dagger_{\text{AB}i}$; $i=1, 2$) when the time-delay circuits work. By assuming that the phase separation dynamics follows the spatiotemporal DNA linker distribution in a mixed DNA droplet, we used a reaction-diffusion model based on the partial differential equations (details in Supplemental Note S1) to estimate the spatiotemporal distribution. The following equations denote the spatiotemporal change of the division triggers $T^\dagger_{\text{AB}i}$ ($i = 1, 2$) controlled by the time-delay circuit:

$$\frac{\partial u_{T^\dagger_{\text{AB}i}}}{\partial t} = D(\mathbf{x})\nabla^2 u_{T^\dagger_{\text{AB}i}} - f_{\text{H-SD}}(\mathbf{u}) + g_{\text{TD}}(\mathbf{u}; c_{\text{ERH}}, u_{\text{R}^\dagger_{\text{AB}i}}^0), \quad (1)$$

$$g_{\text{TD}}(\mathbf{u}; c_{\text{ERH}}, u_{\text{R}^\dagger_{\text{AB}i}}^0) := \frac{k_{\text{cat}}c_{\text{ERH}}u_{iT^\dagger_{\text{AB}i}}}{K_m + u_{iT^\dagger_{\text{AB}i}}} - k_{\text{hRNA}}u_{T^\dagger_{\text{AB}i}}u_{\text{R}^\dagger_{\text{AB}i}}, \quad (2)$$

where u_X is the concentration of molecule “X”; $\mathbf{u} := \{u_{T^\dagger_{\text{AB}i}}, u_{iT^\dagger_{\text{AB}i}}, u_{\text{R}^\dagger_{\text{AB}i}}, \dots\}$ is the vector of concentrations of molecules. The first term in Eq. (1) is the spatial diffusion of $T^\dagger_{\text{AB}i}$; $D(\mathbf{x})$ is the diffusion coefficient depending on the position \mathbf{x} (\mathbf{x} = “inside” or “outside” of A:B-droplet). The second term $-f_{\text{H-SD}}(\mathbf{u})$ denotes the

	consumption of division triggers T_{ABi}^\dagger via hybridization and strand displacement with the linker L_{AB}^\dagger. The third term $g(\mathbf{u}; c_{ERH}, u_{R_{ABi}^\dagger}^0)$ denotes the time-delay circuit reaction composed of the generation and inhibition of T_{ABi}^\dagger, described in Eq. (2) in detail; K_m and k_{cat} are the Michaelis-Menten parameters for the RNase H reaction; c_{ERH} is the total RNase H concentration; $k_{h_{RNA}}$ are the hybridization rates of the division triggers with inhibitor RNAs; $u_{R_{ABi}^\dagger}^0$ ($i = 1, 2$) are the initial concentrations of excess inhibitor RNAs. Thus, the time course of T_{ABi}^\dagger is controlled by two important factors of the time-delay circuit: c_{ERH} and $u_{R_{ABi}^\dagger}^0$.”
4) I believe that $k_{h_{AB}}$ should be a strand displacement rate, rather than a hybridization rate as currently stated.	We are sorry for the confusing explanation. $k_{h_{AB}}$ in the previous manuscript is not wrong because the equation explained only the hybridization process. The strand displacement process is described with another independent equation included in Supplemental Note S1, where we defined the strand displacement rate $k_{SD_{AB}}$ as the reviewer indicated. In the current version of our manuscript, the corresponding equation has been deleted as mentioned above. Therefore, we believe the confusing explanation has been resolved in this version.
5) The patterns in Fig. 3d and e are interesting, and I think deserve further discussion. It seems that the non-delayed linkers in Fig. 3d are eliminated from the outside of the droplet towards the inside, with a propagating front.	We agree with the reviewer’s comments and added the discussion about our numerical results as follows: (p.9-10) “The degradation of L_{AB} occurs from the outside of the droplet towards the inside, while that of L_{AB}^\dagger happens uniformly throughout the droplet. This would be because

This probably occurs because the rate of “reaction” (strand displacement disassembling the linkers) is faster than the rate of diffusion through the droplet, given the high concentration of trigger strands available. Instead, the delayed linkers in Fig. 3e appear to be eliminated uniformly throughout the droplet. This is likely because the rate of linker disassembly is in this case slower compared to the rate of diffusion, given that small amounts of trigger strands are available at any given time due to the action of the RNA blockers. Could the authors provide discussion along these lines if they agree? If not, what could be the cause of the observed difference in behaviour?	the reaction rate is faster than the diffusion rate for L_{AB}, whereas the reaction rate is slower than the diffusion rate for L^{\dagger}_{AB} due to the low amount of active division triggers.”
6) In Fig. 3 f and g: I find it a bit strange that the authors decided to plot the sum of the concentration of the two types of linkers. Why not show their time evolution individually rather than showing the sum? This would be more insightful.	We thank the reviewer for the suggestion. We agree with it. We replaced Figures 3f and 3g with the plot of the individual time evolution of linker concentrations, new Figures 3f-3h as follows: (p.11, Figures 3f-3h)

Additionally, we put the plot of total concentration in Supplemental Figure S1 as follows:

(SI p.17, Supplemental Figure S1)

In the main text, we revised the corresponding sentences:

(p.10)

	“Figure 3f is the time course of DNA linker L_{AB} cleavage, showing L_{AB} cleaves rapidly. Next, we investigated the dependence of the cleavage rate of DNA linkers L_{AB}^\dagger on the total RNase H concentration (c_{ERH}) and the initial concentration of the excess inhibitor RNAs ($u_{R_{AB}i}^0$; $i = 1, 2$) (Figures 3g and 3h). The cleavage rate of the L_{AB}^\dagger becomes slower by decreasing RNase H concentration c_{ERH} (Figure 3g) or increasing RNA concentration $u_{R_{AB}i}^0$ (Figure 3h). By summing them up, the decreasing time courses of total linker concentrations are found to be delayed (Supplemental Figure S1).”
7) In Fig. 4 c and d it is interesting that the division curves are not as sharp as one would expect from simulations. Could the authors comment on this? Could this be because of size polydispersity in the droplets? I.e. do the authors observe a correlation between droplet size and onset time of division?	We think that the difference in the sharpness of r_{div} between the experiments and simulations was probably because of the slow response of B-droplet division against linker cleavage. From our observation, the response of the C-droplet division was faster than that of the B-droplet. The time courses of division ratio r_{div} of C-droplets are sigmoidal, although those of B-droplets do not look like a sigmoidal curve. To answer the reviewer’s questions, we analyzed the size dependence of division time courses for individual droplets as follows.

These graphs show that the division ratio time courses are similar even though the droplet size was different.

To clarify the explanation, we added the sentence as follows:

(p.13)

“The time courses of r_{div} in the experiments were not as sharp as those in the simulation, probably because of the slow response of the B-droplet against linker cleavage.”

8) Data in Fig. 6c become quite noisy at later times, with some sudden jumps and large error bars. Could the authors comment on the origin of these? Do they expect them to produce large errors in the determination of Delta tau?

In experimental conditions that can set up the situation of pathway-controlled division well, the droplets slightly dissolved due to the reaction, which caused noises when the fluorescent image was converted to binary images, resulting in large errors. The errors do not significantly affect the determination of $\Delta\tau$ since the errors were small around $r_{\text{div}} = 0.5$ as shown in Figure 6b.

We have reflected the reviewer's comments by adding the following sentence:

(p.17) "Large errors of r_{div} were observed at the later stage (Figure 6b), which would be because the slight dissolution of droplets made background noises. Since the errors were small around $r_{\text{div}} = 0.5$, the errors did not significantly affect the determination of $\Delta\tau$."

Reviewer 2

In this work, Maruyama et al. present a method for temporally controlling the division of DNA-based liquid-liquid phase separation droplets (DNA droplets). DNA droplets are fascinating because of their capability to mimic natural cellular compartments within synthetic environments. The temporal control of their division is particularly interesting as it has the potential to spatially and temporally control biochemical reactions within artificial cells. Here, the authors achieve temporal control of DNA droplet division using a time-delay circuit, which is based on the degradation of RNA within a RNA-DNA heteroduplexes by RNase H. First, they demonstrate their previously established droplet division method, which employs trigger DNA strands that separate linker structures within the DNA droplets via toehold-mediated strand displacement reactions. Next, they present the principle of their new method, which introduces a time-delay, by using RNA strands that bind to the trigger strands effectively inhibiting the strand displacement reaction. Upon RNA cleavage by RNase H, the trigger strands are released, enabling the separation of the linker structures. Before testing their new method, they perform theoretical simulations based on numerically solving a reaction-diffusion model. The

We thank the reviewer for the positive and constructive comments. We have revised our manuscript based on the reviewer's valuable comments as follows.

simulations qualitatively predict the effect of varying the RNase H, as well as inhibitor RNA concentration, on the division rate of the linker structures. Afterwards, they verify their simulation results by performing experiments of the temporally controlled DNA droplet division using confocal laser scanning microscopy. Furthermore, they show that their method can be upscaled to cleave different linker structures, enabling a droplet division into distinct compartments along different pathways. Finally, they demonstrate an application of their pathway control to realize a molecular comparator for miRNA concentrations. Overall, I enjoyed reading the manuscript, as it is well crafted and easy to follow. The experimental data is well complemented with theoretical considerations. Below are my specific comments regarding the manuscript:	
Major comments:	
1. In the first results section, the authors should state more clearly, that their DNA droplet design as well as division triggering follows their previous work.	We thank the reviewer for the suggestion. We reflected the reviewer’s comment and corrected the sentences as follows: (p.6) “This design is based on our previous study⁴⁷. ...The A·B-droplet started to divide just after adding division triggers. The result agreed with the previous study⁴⁷;”

2. The definition of the droplet division ratio r_{div} (Supplementary Note S3) seems to not match to what is shown in Figures 4c,d and Figure 6c. Given the current definition, for perfectly mixed droplets, the number of pixels having both fluorophores (N_{AB}) should match the number of pixels having the blue fluorophore (N_B), thus the ratio is 1. For perfectly separated droplets, the number of pixels having both fluorophores should be 0, resulting in a ratio of 0. What I think is shown, is $r_{div} = 1 - N_{AB}/N_B$. To resolve this discrepancy, the authors should revise their definition of the droplet division ratio.	We thank the reviewer for pointing out the misdescription. We corrected the description as follows: (SI p.17 Supplemental NoteS3) “The division ratio for B-droplet division, r_{div_B}, was defined as follows: $r_{div_B} = 1 - N_{colc_AB}/N_{total_B}$, where N_{total_B} is the sum of the number of pixels in the droplet regions in the Alexa 405 channel;... .” “In addition, the division ratio for C-droplet division, r_{div_C}, was defined as follows: $r_{div_C} = 1 - N_{colc_AC}/N_{total_C}$, where N_{total_C} is the sum of the number of pixels in the droplet regions in the Cy3 channel and N_{colc_AC} is the sum of the number of pixels that are droplet regions in both the 6-FAM and Cy3 channels.” This writing error does not impact our results at all.
3. There is a discrepancy within the value of the threshold concentration K. In the main text on page 10, a K value of 0.95 is stated to be shown in Figure 3, whereas in supplemental Figure S2, the K value was changed between 0.01, 0.05 and 0.1. It looks like $n=16$ and $K=0.05$ of Figure S2b are shown in Figure 3 h,i. I think this might be due to the threshold concentration once relating to the total linker concentration and once to the linker concentration that can be inhibited? The authors should correct this discrepancy.	We thank the reviewer for pointing out the misdescription, again. We corrected the sentence as follows: (p.10) “Figures 3i and 3j show the time courses of r_{div} when changing RNase H concentration c_{ERH}, and RNA concentration $u_{R^+_{ABi}}^0$ with $K = 0.05$ and $n = 16$ fixed.” This writing error also does not impact our results at all.
4-1. Comparing Figure 3 h,I and Figure 4 c,d, the experimental observations do not align well with the predicted	We received the same comment from Reviewer 1. We think that the difference in the sharpness of r_{div} between the experiments and simulations was probably because of the

sigmoidal shape of the theoretical simulations. The authors should discuss this discrepancy.	slow response of B-droplet division against linker cleavage. From our observation, the response of the C-droplet division was faster than that of the B-droplet. The time courses of division ratio r_{div} of C-droplets are sigmoidal, although those of B-droplets do not look like a sigmoidal curve. To clarify the explanation, we added the sentence as follows: (p.20) “The time courses of r_{div_B} in the experiments were not as sharp as those in the simulation, probably because of the slow response of the B-droplet against linker cleavage.”
4-2. Furthermore, why was a ratio of 90% to 10% chosen for linkers that are not inhibited to linkers that can be inhibited? Would a higher amount of linkers that can be inhibited lead to a more sigmoidal shape as predicted by the simulations?	In this study, we just set the linker concentration the same as in Sato’s paper. We used a higher amount of inhibited linkers in the experiments in Figure 6b. In Figure 6b, 100% of the inhibited linker was used. However, the curve of r_{div_B} did not get very sharp. This result suggests that the ratio of linkers was not so related to the sharpness of the r_{div} time course. As we mentioned above (the response to the reviewer’s comment 4-1), we think the sharpness of r_{div} time course is probably involved in the response of division to the linker cleavage. We believe that the sentences mentioned in the response to 4-1 would resolve this question.

5. The authors should quantify their observation for the control of the droplet division pathway (Figure 5c,d) similar to Figure 6c.

We thank the reviewer for the suggestion.

We agree with the suggestion and put the quantitative results as Figures 5f and 5g and the explanation as follows:

(p.15, Figure 5)

	(pp.14-15) “Furthermore, we quantified the time courses of the division ratios of B- ($r_{\text{div_B}}$) and C- ($r_{\text{div_C}}$) droplets using the image processing method shown in Supplementary Note S3. The results showed that the increase of $r_{\text{div_B}}$ was slower than $r_{\text{div_C}}$ in Pathway 1 (Figure 5f), while that of $r_{\text{div_C}}$ was slower than $r_{\text{div_B}}$ in Pathway 2 (Figure 5g).” Furthermore, we added the sentence in the caption of Figure 6 to indicate that Figure 6b (i) and (v) are identical to Figures 5f and 5g, respectively, as follows: (p.20) “The plots in conditions (i) and (v) are identical to those in Figures 5f and 5g, respectively.”
6. Figure 6c shows much better division delays compared to Figure 4. The authors should provide supplementary videos showing some of these reactions.	We added the movies for the conditions (ii)-(iv) in Figure 6b as Supplemental Movies S8-10. The movies for (i) and (v) are the same as the Supplemental Movies S6 and S7.
Minor comments:	
1. The supplementary videos provide valuable visual insight into the temporal control of the DNA droplets. Unfortunately, some of the traces exhibit noticeable drift or abrupt jumps along the time trajectory, making it hard to follow single droplets. Since the videos are already processed within FIJI, I suggest considering the implementation of a simple drift correction using a plugin such as StackRegJ, Fast4DReg, or the manual drift correction plugin. This could enhance the overall quality of the presented data.	We thank the reviewer for the fruitful suggestion. We analyzed our data using StackRegJ as suggested. We, however, found that this method with StackRegJ could not be applied to our study because some droplets were framed out due to the tilting of the analyzed image, although the entire image was necessary for analysis in this study. We thank the reviewer again for the valuable suggestion.

2. Overall, the experiments and conclusions are well and logically described and clearly understandable. However, the manuscript would benefit from improving the grammar of the text.	We checked and corrected the grammar of the text.
3. Page 9 line 5: the period should be behind the “”, to read: ”X”.	We corrected the relevant mistake.
4. The term “decreasing rate” is used to describe the decrease in concentration of the Linkers. It becomes confusing when used together with decreasing and increasing concentrations. I suggest to use more specific terms such as “LAB cleavage rate” or even better “LAB cleavage kinetics” since the simulations provide a kinetics rather than a rate.	We agree with the reviewer’s comments and changed “decreasing rate” to “cleavage rate” as follows: (p.10) Figure 3f is the time course of DNA linker L_{AB} cleavage, showing L_{AB} cleaves rapidly. Next, we investigated the dependence of the cleavage rate of DNA linkers L_{AB}^\dagger on the total RNase H concentration (c_{ERH}) and the initial concentration of the excess inhibitor RNAs ($u_{R^\dagger_{ABi}}^0$; $i = 1, 2$) (Figures 3g and 3h). The cleavage rate of the L_{AB}^\dagger becomes slower by decreasing RNase H concentration c_{ERH} (Figure 3g) or increasing RNA concentration $u_{R^\dagger_{ABi}}^0$ (Figure 3h). By summing them up, the decreasing time courses of total linker concentrations are found to be delayed (Supplemental Figure S1).
5. I also suggest to avoid the usage of formula symbols when discussing concentrations and other parameters in the text (e.g. decreasing c_{ERH} or increasing u_{RABi}) wherever this is possible. This makes it easier to follow the text.	We added the word “RNase H concentration” and “RNA concentration” to follow the text easier, for example: (p.10) The cleavage rate of the L_{AB}^\dagger becomes slower by decreasing RNase H concentration c_{ERH} (Figure 3g) or increasing RNA concentration $u_{R^\dagger_{ABi}}^0$ (Figure 3h).

Reviewer 3

Maruyama et al. present a method to achieve division dynamics in artificial DNA-based droplets by using multiple time-delayed triggers controlled by out-of-equilibrium chemical reactions. First, they generate binary mixed DNA droplets by combining two Y-shaped DNA nanostructures, YA and YB, connected by 6-branched DNA linkers (A:B droplets). Division is achieved by cleavage of the linkers through hybridization with division-triggering DNAs. The main goal of the project is to develop a time-delay circuit with a reaction that prevents division by hybridization with excess single-stranded inhibitor RNAs. This leads to a temporal delay of linker cleavage and consequently the division of DNA droplets by RNA degradation with ribonuclease H (RNase H). Finally, they explored an alternative method for generating C-A-B droplets using two different types of DNA cleavage linkers, which enabled the construction of a molecular comparator for miRNA concentrations. The proposed approach is clever in itself and offers an interesting opportunity to specifically divide DNA condensates. The manuscript and the SI are generally well organized; the claims about the timed division dynamics are supported by the data, and by what one would predict based on previous work by the authors and others.

We thank the reviewer for the overall positive comments and constructive comments. We have revised our manuscript as follows.

1) While the use of RNase H for the temporal control of DNA-based LLPS droplets is innovative, the overall novelty of the concept is perhaps limited. First, the authors already used enzymes cleaving 6 arm nanostars to demonstrate droplet division (Sci Adv 2020, 6 (23), eaba3471). The use of DNA triggers is its natural extension, given recent work showing how strand displacement/invasion can be used to modify the properties of DNA nanostar-based condensates (Science Advances 8.41 (2022): eabj1771 and Nature Communications 15.1 (2024): 1915). Previous work has also reported that the timing of DNA-based reactions and assemblies can be tuned through the use of RNase H (JACS 145.38 (2023): 20968-20974, JACS 143.48 (2021): 20296-20301). In addition, the use of pathway control for molecular computation and comparator of miRNA concentrations may not be particularly groundbreaking, as the same group has shown in 2022 (Adv. Funct. Mater. 2022, 2202322) that detection of miRNA is possible through the development of DNA-responsive droplets.

We thank the reviewer for the comments.

We believe that our study has the novelty in that we achieved the timing control of synthetic LLPS droplet dynamics by coupling it with chemical reactions and demonstrated complex dynamics and applications using the temporal control of droplet dynamics.

Temporally controlling the dynamics of cellular compartments by chemical reactions is essential feature of living cells. Therefore, in synthetic biology, the achievement of temporal control of dynamics of synthetic LLPS droplet by chemical reactions are desired. As the reviewer mentioned, the assembly/disassembly and division of DNA droplets have been already reported, but the temporal control of such dynamics was not achieved yet.

Furthermore, regarding pathway control, the previous study (*Adv. Funct. Mater.* 2022, 2202322) focused on the pre-division state and the final post-division state because the purpose was to detect miRNA sequences; i.e., the dynamics during division was not controlled. In contrast, in this study, we focused on the dynamics during the division process and achieved the control of the division pathway via temporal control of division dynamics (Figure 5d, 5e).

Also, from the view of the detection of miRNA sequences using DNA droplet, the previous research just evaluated the presence or absence of miRNA sequences. In contrast, this research enabled the comparison of the concentration of miRNA sequences in addition to the presence or absence of them.

	To clarify the achievement of this study, we added the following sentences in the Conclusion part: (p.20) “We achieved not only the detection of the presence/absence of miRNA sequences but also the comparison of the concentrations of miRNA sequences, which may be applied to a diagnosis based on the expressed miRNA concentrations.”
2-1) The model is a nice addition, although it is quite complex and therefore difficult to follow - it is well formulated as far as I can tell. I was initially a bit confused about the use of a purely diffusive term in the PDE, instead of a Cahn Hilliard term that is appropriate for phase separation, but I understand that since the total concentration of monomers is actually not changing, and there is no phase transition, this simplified approach is ok.	We thank the reviewer for pointing out the important point. We are sorry for confusing the reviewer. First of all, we intended to model the linker DNA decrease depending on the concentrations of RNase H and the inhibitor RNAs when the time-delay circuits work, but did not model the phase separation dynamics. That is why we used the simple reaction-diffusion model. In our assumption, the phase separation occurs by following the dynamics of the linker DNA decrease. As the reviewer mentioned, we should use the Cahn-Hilliard model if we explicitly model the phase separation process. To avoid confusion, we revised the explanations for our model (purpose, limitations of the model, comparison with the Cahn-Hilliard model with a citation (Nature Communications 15, 1915, 2024)): (p.9)

	“First, we numerically investigated the dependence of the cleaving rate of the DNA linker L_{AB}^\dagger on the concentrations of RNase H and the inhibitor RNAs (R_{ABi}^\dagger; $i=1, 2$) when the time-delay circuits work. Since the phase separation dynamics is determined by the spatiotemporal DNA linker distribution in a mixed DNA droplet, we used a reaction-diffusion model based on the partial differential equations (details in Supplemental Note S1) to estimate the spatiotemporal distribution.” (p.20) “Although the current simulation model focused on the spatiotemporal distribution of linker DNAs to estimate the time-delay circuit behavior, the model would be extended to a model explicitly considering the phase separation process by adding the Cahn-Hilliard term⁵⁸.”
2-2) A question I have is the choice of a very high Hill coefficient ($n=16$) for modeling division ratio as a function of the trigger, which I think is why the authors get very steep division curves in Fig. 3h and i. In these plots, changing RNase H level or inhibitor concentration creates a finite delay without altering the slope of the curve. In contrast, experiments in Fig. 4 c and d show that if RNase H and inhibitor level are changed, there is no clear delay while the slope of the division rate changes. Even in the fastest case (orange curves), the slope of the simulations is much larger when compared to the data. I think the Hill coefficient that was chosen is too large; if I understand correctly, one needs two copies	As the reviewer mentioned, $n = 2$ is enough to model the cooperativity. By comparing the simulation time courses (new Supplemental Figure S2) and the experimental ones (Figures 4c and 4d), we found that the trend of the time courses was consistent with each other in all cases $n = 2, 4, 8, 16,$ and 32. However, the trend of the simulation results (Figure 6d) was more consistent with the experimental results (Figure 6b) when $n \geq 16$. That is why we chose $n = 16$ for the numerical models in Figures 3 and 6 in common. Anyway, the choice of n values did not significantly affect the conclusion that the order of division changed depending on Δc.

of trigger to split one nanostar, so I would expect the Hill coefficient to be 2. Why did the authors choose $n=16$? Perhaps there are aspects of how data were normalized that I do not understand, but the most important output of the model right now behaves quite differently from the data.

To compare the dependence of results on n values, we added numerical simulation results for $n = 2$ and 4 in Supplemental Figures S2 and S8 as follows:

(SI p.18, Supplemental Figure S2)

Supplemental Figure S2. Time course of the division ratio r_{div} in the reaction-diffusion simulation. Time courses of r_{div} at $n = 2$ (a), 4 (b), 8 (c), 16 (d), and 32 (e). For each condition, the value of K was changed to 0.01, 0.05, and 0.10. In all

results, we consistently observed the trend of the increasing rate of r_{div} become slow when increasing $u_{\text{R}^{\dagger}\text{AB}i}^0$ ($i=1, 2$) or decreasing c_{ERH} .

(SI p.24, Supplemental Figure S8)

Supplemental Figure S8. Time difference $\Delta\tau$ at each of five RNA conditions in the reaction-diffusion simulation. $\Delta\tilde{c} =$ (i) 1.25, (ii) 0.5, (iii) -0.5, (iv) -1.0, and (v) -1.25. $k_{\text{hAB}}/k_{\text{hAC}} = k_{\text{SDAB}}/k_{\text{SDAC}} = 0.1$ and $n = 2$ (a), 4 (b), 8 (c), 16 (d), and 32 (e).

3) I find the overall narrative of the paper to be somewhat controversial. I think that when DNA triggers are added, there is no active, autonomous process happening as the authors claim.

We agree with the reviewer's idea about the explanation of this system. As the reviewer mentioned, the overall thermodynamic landscape of the system is changed by the new DNA, and the ensemble of strands just moves to a new equilibrium. This

The overall thermodynamic landscape of the system is changed by the new DNA, and the ensemble of strands just moves to a new equilibrium. I don't understand how this is active, or how any fuel is used to maintain the system out of equilibrium. The old equilibrium no longer exists given how the system is designed. My concept of a non-equilibrium system is one where a fuel molecule maintains the system to an otherwise energetically unfavorable equilibrium state; when fuel is removed from the system, it will relax to its original, energetically stable equilibrium. In this sense, triggers are not a fuel molecule, they are just new components that change the equilibrium to a new resting state that didn't exist before.

Conceptually, I think that if the division trigger strands were RNA, degraded by RNase H (which should be present from the start), then one could say that as long as RNA fuel is present, then the system is maintained in an out-of-equilibrium state (split A/B droplets). As soon as RNase H is done degrading the RNA fuel, then the system would revert to the original equilibrium (mixed A/B droplets).

transition process is called a relaxation process, which is one of non-equilibrium processes. However, the degree of non-equilibrium of the relaxation process is not so high. Here, the trigger molecules are used to make an unstable (energetically higher) state (thermodynamic landscape of the system is changed) and the free energy of the trigger molecules was used and lost through the process of relaxation to the new equilibrium.

The non-equilibrium process maintained using sustained fuel molecules, which the reviewer indicated, is a higher-degree nonequilibrium process. We agree with the reviewer's concept of an out-of-equilibrium state. In this study, we did not aim to maintain an out-of-equilibrium state. In the future study, we would like to achieve an out-of-equilibrium state that the reviewer mentioned.

The confusion in the word of 'non-equilibrium' would have risen because of the misleading illustration in Figure 2f. Reviewer 1 also pointed out the same point. Thus, we reflected the comments and remade the diagram of non-equilibrium process in Figure 2f. The following correction would also resolve the confusion the reviewer 3 pointed out.

We changed the Figure 1 caption: “Description of the A·B-droplet division dynamics based on reaction landscapes. The ssDNA division triggers work as “fuel” molecules, which change the reaction landscape from a single-minimum shape to a double-minimum shape with three transition steps. (i) A·B-droplet with ssDNA fuels outside, where the DNA linker is not cleaved yet. (ii) A·B-droplet with ssDNA fuels inside, where DNA linker is cleaved but the DNA droplet is not divided yet; (iii) A- and B-droplets are separated after the spinodal decomposition. ΔG_{CIV} and ΔG_{PS} are Gibbs free energy changes for the linker cleavage reaction and the phase separation, respectively.”

4) Could the division reactions be reversed and how? Can the original conformation of the mixed droplets be regained starting from split droplets?	The regain of original mixed state from the divided state may be achieved if the trigger connected with the linker is detached by a reaction such as a strand displacement reaction. However, we did not perform such an experiment because we did not focus on the reverse reaction from divided droplets into the original mixed state in this study. We added a sentence related to this comment as useful discussion as follows: (p.21) In future, the control of chemical reactions via the physical dynamics of DNA droplets and the reversible control of DNA droplet dynamics should be explored.
5-1) I do not understand how the comparator circuit works, in the main paper there is not enough detail. Can the authors please clarify with a schematic how the “triangle” in Fig. 6a works?	We are sorry for the lack of explanations of the comparator. The “triangle” in Figure 6a is just a symbol for a comparator generally used in electric circuits. In general, a comparator in electric circuits outputs a voltage depending on the result of the comparison of the voltages of two inputs. Inspired by such a comparator of electric circuits, we developed a comparator that can compare the input RNA concentrations. To clarify the meaning of the triangle symbol, we add the following explanation in Figure 6 caption: “The triangle is a symbol for a comparator element.”

In addition, to provide detailed information of how the comparator works, we revised the schematic illustration in Figure 6a and added detailed explanations in Figure 6 caption.

Figure 6. Application of pathway control to a molecular comparator for miRNA concentrations. (a) Concept of a molecular comparator of miRNA

concentrations. The triangle is a symbol for a comparator element. miRNAs miR-6875-5p and miR-4634 were used for input 1 for the comparator; miR-1246 and miR-1307-3p were used for the input 2. The output is the selection of the droplet division pathway, which changes depending on the difference between two normalized initial total concentrations of inhibitor RNAs $\tilde{c}_{AB} = u_{R^{\dagger}ABi}^{\text{tot}}/u_{T^{\dagger}ABi}^{\text{tot}}$ and $\tilde{c}_{AC} = u_{R^{\dagger}ACi}^{\text{tot}}/u_{T^{\dagger}ACi}^{\text{tot}}$ ($i=1,2$), where the input initial total RNA concentrations are defined as $u_{R^{\dagger}AB1}^{\text{tot}} = [\text{miR-6875-5p}]$, $u_{R^{\dagger}AB2}^{\text{tot}} = [\text{miR-4634}]$, and $u_{R^{\dagger}AB1}^{\text{tot}} = u_{R^{\dagger}AB2}^{\text{tot}}$; $u_{R^{\dagger}AC1}^{\text{tot}} = [\text{miR-1246}]$, $u_{R^{\dagger}AC2}^{\text{tot}} = [\text{miR-1307-3p}]$, and $u_{R^{\dagger}AC1}^{\text{tot}} = u_{R^{\dagger}AC2}^{\text{tot}}$. Pathway 1 (C-droplet is divided first) is selected if $\Delta\tilde{c} = \tilde{c}_{AB} - \tilde{c}_{AC} > \sigma$; Pathway 2 (B-droplet is divided first) is selected in the other case. σ is an offset of comparison; if $\sigma = 0$, the comparator simply computes whether $\tilde{c}_{AB} > \tilde{c}_{AC}$ or not. This comparison is achieved by the two time-delay circuits. (b) ...

In addition, to help the understanding of the comparator, we added a simple case when offset $\sigma = 0$ as Supplemental Figure S3 in SI:
(SI p.19, Supplemental Figure S3)

Supplemental Figure S3. Schematic illustration of concentration comparator of miRNA sequences using pathway-controlled division of C·A·B-droplet. For simplicity, this scheme is drawn when the offset $\sigma = 0$.

5-2) What part of the nanostar/linker strands had to be redesigned, if any, to become responsive to the miRNA sequences?

For the concentration comparator experiment, we did not redesign the nanostars and linkers. We used the same nanostars and linkers as those for the pathway-controlled division experiment shown in Figure 5b.

To reflect the reviewer's comment, we added the explanation as follows:

	(p.16) “In the experiments, we used the same DNA nanostructures as those in Figure 5b.”
(Remarks on code availability)	
The code is provided in python, which I normally don't use. I have not tried to run it myself. It could be commented a bit more. Also the authors should make sure that all comments are in both Japanese and English. Right now they are mostly in Japanese.	To reflect the reviewer's comment, we added more comments throughout the code to improve its readability and understandability. We replaced all Japanese comments with English ones.

Reviewer 4

I co-reviewed this manuscript with one of the reviewers who provided the listed reports. This is part of the Nature Communications initiative to facilitate training in peer review and to provide appropriate recognition for Early Career Researchers who co-review manuscripts.	We thank the reviewer for reviewing our manuscript.
---	--

Other minor corrections

	We found errors in writing about the definition of \tilde{c}_{AB} and \tilde{c}_{AC}. We corrected them as below. These are just errors in writing in the texts. By these changes, the resultant data of simulations and experiments have not changed. (p.12) Normalized initial total concentration of inhibitor RNA is defined as $\tilde{c}_{AB} = u_{R^{\dagger}ABi}^{\text{tot}}/u_{T^{\dagger}ABi}^{\text{tot}}$ ($i=1,2$), where $u_{R^{\dagger}ABi}^{\text{tot}} = u_{R^{\dagger}ABi}^0 + u_{iT^{\dagger}ABi}^0$ is the initial total
--	---

concentration of excess and hybridized inhibitor RNAs, and $u_{T^{\dagger}ABi}^{\text{tot}} = u_{T^{\dagger}ABi}^0 + u_{iT^{\dagger}ABi}^0$ is the initial total concentration of active and inhibited triggers.

(p.16)

two normalized initial total concentrations of inhibitor RNAs \tilde{c}_{AB} and \tilde{c}_{AC} ($\tilde{c}_{AB} = u_{R^{\dagger}ABi}^{\text{tot}}/u_{T^{\dagger}ABi}^{\text{tot}}$ and $\tilde{c}_{AC} = u_{R^{\dagger}ACi}^{\text{tot}}/u_{T^{\dagger}ACi}^{\text{tot}}$ ($i=1,2$) are defined in the same way (see Figure 3 caption)).

(p.19, Figure 6 caption)

In this experiment, the input initial total RNA concentrations: $u_{R^{\dagger}AB1}^{\text{tot}} = [\text{miR-6875-5p}]$, $u_{R^{\dagger}AB2}^{\text{tot}} = [\text{miR-4634}]$, and $u_{R^{\dagger}AB1}^{\text{tot}} = u_{R^{\dagger}AB2}^{\text{tot}}$; $u_{R^{\dagger}AC1}^{\text{tot}} = [\text{miR-1246}]$, $u_{R^{\dagger}AC2}^{\text{tot}} = [\text{miR-1307-3p}]$, and $u_{R^{\dagger}AC1}^{\text{tot}} = u_{R^{\dagger}AC2}^{\text{tot}}$.

(p.20, Figure 6 caption)

two normalized initial total concentrations of inhibitor RNAs $\tilde{c}_{AB} = u_{R^{\dagger}ABi}^{\text{tot}}/u_{T^{\dagger}ABi}^{\text{tot}}$ and $\tilde{c}_{AC} = u_{R^{\dagger}ACi}^{\text{tot}}/u_{T^{\dagger}ACi}^{\text{tot}}$ ($i=1,2$).

(SI p.3)

Normalized initial total concentration of inhibitor RNA is defined as $\tilde{c}_{AB} = u_{R^{\dagger}ABi}^{\text{tot}}/u_{T^{\dagger}ABi}^{\text{tot}}$ ($i=1,2$), where $u_{R^{\dagger}ABi}^{\text{tot}}$ is the concentration of $R^{\dagger}ABi$, and $u_{T^{\dagger}ABi}^{\text{tot}}$ is the concentration of $T^{\dagger}ABi$.

(SI p.5)

Normalized initial total concentration of inhibitor RNA \tilde{c}_{AB} is defined as $\tilde{c}_{AB} = u_{R^{\dagger}ABi}^{\text{tot}}/u_{T^{\dagger}ABi}^{\text{tot}}$ ($i=1,2$), where $u_{R^{\dagger}ABi}^{\text{tot}}$ is the concentration of $R^{\dagger}ABi$, and $u_{T^{\dagger}ABi}^{\text{tot}}$ is the concentration of $T^{\dagger}ABi$. Normalized initial total concentration of inhibitor RNA \tilde{c}_{AC} is defined as $\tilde{c}_{AC} = u_{R^{\dagger}ACi}^{\text{tot}}/u_{T^{\dagger}ACi}^{\text{tot}}$ ($i=1,2$), where $u_{R^{\dagger}ACi}^{\text{tot}}$ is the concentration of $R^{\dagger}ACi$, and $u_{T^{\dagger}ACi}^{\text{tot}}$ is the concentration of $T^{\dagger}ACi$.

(SI p.12)

Normalized initial total concentration of inhibitor RNA $\tilde{c}_{AB} = u_{R^{\dagger}ABi}^{\text{tot}}/u_{T^{\dagger}ABi}^{\text{tot}} = (u_{R^{\dagger}ABi}^0 + u_{iT^{\dagger}ABi}^0)/(u_{T^{\dagger}ABi}^0 + u_{iT^{\dagger}ABi}^0)$ is defined.

(SI p.15)

Normalized initial total concentration of inhibitor RNA $\tilde{c}_{AB} = u_{R^{\dagger}ABi}^{\text{tot}}/u_{T^{\dagger}ABi}^{\text{tot}} = (u_{R^{\dagger}ABi}^0 + u_{iT^{\dagger}ABi}^0)/(u_{T^{\dagger}ABi}^0 + u_{iT^{\dagger}ABi}^0)$, where $u_{R^{\dagger}ABi}^{\text{tot}} = u_{R^{\dagger}ABi}^0 + u_{iT^{\dagger}ABi}^0$ is the initial total concentration of excess and hybridized inhibitor RNAs, and $u_{T^{\dagger}ABi}^{\text{tot}} = u_{T^{\dagger}ABi}^0 + u_{iT^{\dagger}ABi}^0$ is the initial total concentration of active and inhibited triggers. Similarly, $\tilde{c}_{AC} = u_{R^{\dagger}ACi}^{\text{tot}}/u_{T^{\dagger}ACi}^{\text{tot}} = (u_{R^{\dagger}ACi}^0 + u_{iT^{\dagger}ACi}^0)/(u_{T^{\dagger}ACi}^0 + u_{iT^{\dagger}ACi}^0)$.

	Statistical information was added. Figures 4c and 4d: Three repeated experiments in each condition are shown with the same color. Figures 5f and 5g: Error bars: standard errors of more than 4 observations. Figure 6b: Error bars: standard errors of 3 observations. Figure 6c: Error bars: standard errors of more than 4 observations.
	The third affiliation was changed (the name of the research center was changed): ³ Research Center for Autonomous Systems Materialogy (ASMat), Institute of Innovative Research, Tokyo Institute of Technology, 4259 Nagatsuta-cho, Midori-ku, Yokohama, Kanagawa 226-8501, Japan
	“Data availability”, “Code availability”, “Author information”, and “Ethics declaration” parts were added as follows. Data availability The dataset of the main figures generated in this study is provided in the Supplementary Information and Supplementary Videos.

Code availability

The source codes for numerical simulations are provided through GitHub (<https://github.com/takinouelab/MaruyamaTakinoue2024>).

Author information

Department of Life Science and Technology, Tokyo Institute of Technology, 4259 Nagatsuta-cho, Midori-ku, Yokohama, Kanagawa 226-8501, Japan

Tomoya Maruyama, Jing Gong, Masahiro Takinoue

Department of Computer Science, Tokyo Institute of Technology, 4259 Nagatsuta-cho, Midori-ku, Yokohama, Kanagawa 226-8501, Japan

Masahiro Takinoue

Research Center for Autonomous Systems Materialogy (ASMat), Institute of Innovative Research, Tokyo Institute of Technology, 4259 Nagatsuta-cho, Midori-ku, Yokohama, Kanagawa 226-8501, Japan

Masahiro Takinoue

Contributions

M.T. provided the original concept. T.M. and M.T. performed all experiments. T.M., J.G., and M.T. design experiments. T.M. and M.T. wrote and revised the manuscript.

Corresponding author

Correspondence to Masahiro Takinoue.

Ethics declaration

Competing interests

The authors declare no competing interests.

REVIEWER COMMENTS

Reviewer #1 (Remarks to the Author):

The authors have done a very good job addressing my comments. I am very happy to recommend publication.

Reviewer #2 (Remarks to the Author):

In the revised version the authors have addressed all raised points, such that publication of the manuscript can be recommended.

Reviewer #3 (Remarks to the Author):

The authors made efforts to address my comments in the revised manuscript.

I still have the same concerns about the overall novelty. The authors respond that “assembly/disassembly and division of DNA droplets have been already reported, but the temporal control of such dynamics was not achieved yet” - I disagree with this statement in general, in the sense that control over the dynamics of assembly and disassembly has been shown before, by changing the length of toeholds or concentration of triggers. The same thing is done here by changing the amount of RNase H (Fig. 4). The demonstration of delayed onset of division due to RNase H degradation has not been shown yet for DNA condensates, as far as I know. However, the basic principle used here is sequestration of the trigger through excess inhibitors, which is a well-known trick in the DNA nanotech field.

I also disagree with the author's response to my comments regarding the non-equilibrium, “active” nature of the system, as well as the sketch of the energy landscape.

There are two issues with how the system in Fig. 2 is described: 1) The system is not active in any way, since there is no energy consumption when the trigger hybridizes to the nanostar linkers; 2)

There is no energy barrier between the non-divided and the divided “states”. Adding the trigger changes the energy landscape so a new equilibrium emerges, and the system reaches it spontaneously. Why would there be a barrier? The trigger hybridizes with the linking strands, so there is no longer a linker for nanostars A and B, and they self-segregate in distinct droplets.

Finally, the system is already in the spinodal region prior to adding trigger, so it is confusing to have it marked near the right well of the energy landscape but not the left one. (At any rate, there should not be two distinct wells in the bottom sketch.)

As for the explanation of how the comparator works - I am sorry but I still don't have enough details to understand how the gray triangle in Fig. 6a is supposed to work. What is σ ? What are the DNA strands associated with the comparator and how do they interact? Also SI Fig. S3 is identical to Fig. 6a and it does not add new information, so why is it included? Is this perhaps a mistake and a different figure S3 was meant to be included?

Overall, the value of the simulations is somewhat minor. While I really appreciate the effort to be rigorous, the complexity of the model does not provide the advantages one would expect when compared to a more minimalistic, intuitive model. First, the complex model does not reproduce data in an accurate way; second, it actually loses connection with reality given the extremely high Hill coefficients adopted.

In conclusion, I think this is solid work, the results are visually appealing, and it definitely deserves publication - the approach is of interest for people working in the field of DNA condensates. I don't think the best venue for publication is a general audience journal such as this one.

Reviewer #4 (Remarks to the Author):

Reviewer 1

The authors have done a very good job addressing my comments. I am very happy to recommend publication.	We thank the reviewer for reviewing our manuscript again and giving us the recommendation comment.
--	---

Reviewer 2

In the revised version the authors have addressed all raised points, such that publication of the manuscript can be recommended.	We thank the reviewer for reviewing our manuscript again and giving us the recommendation comment.
---	---

Reviewer 3

The authors made efforts to address my comments in the revised manuscript.	We thank the reviewer for reviewing our manuscript again.
[1] I still have the same concerns about the overall novelty. The authors respond that “assembly/disassembly and division of DNA droplets have been already reported, but the temporal control of such dynamics was not achieved yet” - I disagree with this statement in general, in the sense that control over the dynamics of assembly and disassembly has been shown before, by changing the length of toeholds or concentration of triggers. The same thing is done here by changing the amount of RNase H (Fig. 4). The demonstration of delayed onset of division due to RNase H degradation has not been shown yet for DNA condensates, as far as I know. However, the basic principle used here is sequestration of the trigger through excess inhibitors, which is a well-known trick in the DNA nanotech field.	About the description of the temporal control, we have not intended to claim that our temporal control of DNA droplet division is the first case of the temporal control of DNA reactions in general. What we would like to report is just that the pathway of DNA droplet division is controlled by RNase H-based reactions as the reviewer mentioned “delayed onset of division due to RNase H degradation has not been shown yet for DNA condensates.” We, of course, know that the RNase H-based trick has been frequently utilized in the DNA nanotechnology field. Therefore, we have mentioned the previous studies using the RNase H-based trick with citations in the introduction part. One novelty of this study would be the demonstration that the RNase H-based trick could be applied to the control of the phase separation dynamics as well as the other DNA reaction networks in a bulk solution. The other novelty would be the construction of a computation element, a comparator, using the control of the phase separation dynamics with the RNase H-based trick. We guess our descriptions on these points were unclear. We added the following description to make the points clearer. (p.23) Based on these results, we revealed that the RNase H-based strategy could be applied to the control of the phase separation dynamics as well as the other DNA nanotechnologies in a bulk solution.

[2-1] I also disagree with the author's response to my comments regarding the non-equilibrium, "active" nature of the system, as well as the sketch of the energy landscape. There are two issues with how the system in Fig. 2 is described: 1

1) The system is not active in any way, since there is no energy consumption when the trigger hybridizes to the nanostar linkers;

We agree with the reviewer's comment that our system is not active because there is no energy consumption. Since we did not supply sustained chemical energy for the phase separation control as the reviewer mentioned, we did not intend to claim that our system is an active system (a far-from-equilibrium system) at all. However, the usage of some terms may have confused the reviewer: the first one would be 'non-equilibrium chemical reaction'; the second one may be 'active division triggers'; and the other one would be 'fuel'.

In the previous manuscript, we used the term 'non-equilibrium chemical reaction' for a transient non-equilibrium relaxation process to reach an equilibrium state, in which the RNA is degraded by RNase H. Since this reaction is not a process using chemical reactions far from equilibrium as the reviewer pointed out, we changed the description to just 'chemical reaction.' Moreover, to define that this study focuses on only a transient non-equilibrium relaxation process, we added the definition in the Introduction as follows:

In the present study, we demonstrated the timing-controlled division dynamics of DNA droplet-based artificial cells by coupling them with **chemical reactions exhibiting a transient non-equilibrium relaxation process**, resulting in the pathway control of artificial cell division (Figure 1).

In addition, to clarify the unachieved point ('active' system), we added the following description in the Conclusion:

The present study demonstrated that **chemical reactions** could control DNA droplet dynamics such as droplet division. **However, since the coupled chemical reactions were only a transient non-equilibrium relaxation process, far-from-equilibrium chemical reactions with sustained chemical energy supplies are**

required to achieve truly active systems.

We used the term ‘active’ as the meaning of ‘available’ because the ‘division triggers’ are released from the inhibited state (DNA-RNA duplex). Since the ‘active’ may confuse that our system is an ‘active’ system, we changed the word to ‘released division triggers.’

Regarding the ‘fuel’, we used the word ‘fuel’ with the landscape change since the pioneering research on DNA hybridization-based state transition (Yurke, B.; Turberfield, A. J.; Mills, A. P., Jr; Simmel, F. C.; Neumann, J. L. *A DNA-Fuelled Molecular Machine Made of DNA. Nature* 2000, 406 (6796), 605–608.) used the word ‘fuel.’ However, we guess that the word ‘fuel’ with the landscape in Figure 2 causes the misunderstanding that our system is active, having changed the use of the term. In addition, we simplified Figure 2f using the single-minimum landscape and the description in the caption. This figure change was related to the reviewer’s comment below, too.

(f) Description of the A·B-droplet division dynamics based on reaction landscapes. The ssDNA division triggers change the reaction landscape from a single-minimum shape: (i) A·B-droplet with ssDNA triggers but the A·B-droplet is not divided yet; (ii) A- and B-droplets are divided through the spinodal decomposition. ΔG_{CIV} and

ΔG_{PS} are Gibbs free energy changes for the linker cleavage reaction and the phase separation, respectively.

[2-2] 2) There is no energy barrier between the non-divided and the divided “states”. Adding the trigger changes the energy landscape so a new equilibrium emerges, and the system reaches it spontaneously. Why would there be a barrier? The trigger hybridizes with the linking strands, so there is no longer a linker for nanostars A and B, and they self-segregate in distinct droplets. Finally, the system is already in the spinodal region prior to adding trigger, so it is confusing to have it marked near the right well of the energy landscape but not the left one. (At any rate, there should not be two distinct wells in the bottom sketch.)

We considered that the energy barrier in our free energy landscape was related to the entropic cost for the trigger penetration and the first contact of the trigger to the linker, which is also described as the process ‘A’ to ‘B’ in Figure 3 of the previous paper below (Srinivas, N.; Ouldrige, T. E.; Sulc, P.; Schaeffer, J. M.; Yurke, B.; Louis, A. A.; Doye, J. P. K.; Winfree, E. On the Biophysics and Kinetics of Toehold-Mediated DNA Strand Displacement. *Nucleic Acids Res.* 2013, 41 (22), 10641–10658):

Figure 3. Free energy landscape of the IEL at 25°C for a six base toehold. States A–F and the sawtooth amplitude (ΔG_s) and plateau height (ΔG_p) parameters are described in the text. $\Delta G_s = 2.6$ kcal/mol and $\Delta G_p = 1.2$ kcal/mol are used for illustration.

However, the description of the detailed process may confuse the readers. Therefore, we simplified our Figure 2f using the single-minimum landscape and the description in the caption as follows.

(Previous version)

(f) Description of the A·B-droplet division dynamics based on reaction landscapes. The ssDNA division triggers work as “fuel” molecules, which change the reaction landscape from a single-minimum shape to a double-minimum shape with three transition steps: (i) A·B-droplet with ssDNA fuels outside, where the DNA linker is not cleaved yet; (ii) A·B-droplet with ssDNA fuels inside, where DNA linker is cleaved but the A·B-droplet is not divided yet; (iii) A- and B-droplets are divided through the spinodal decomposition. ΔG_{CIV} and ΔG_{PS} are Gibbs free energy changes for the linker cleavage reaction and the phase separation, respectively.

(Current revised version)

(f) Description of the A·B-droplet division dynamics based on reaction landscapes. The ssDNA division triggers change the reaction landscape from a single-minimum shape: (i) A·B-droplet with ssDNA triggers but the A·B-droplet is not divided yet; (ii) A- and B-droplets are divided through the spinodal decomposition. ΔG_{Clv} and ΔG_{Ps} are Gibbs free energy changes for the linker cleavage reaction and the phase separation, respectively.

[3] As for the explanation of how the comparator works - I am sorry but I still don't have enough details to understand how the gray triangle in Fig. 6a is supposed to work. What is sigma? What are the DNA strands associated with the comparator and how do they interact? Also SI Fig. S3 is identical to Fig. 6a and it does not add new information, so why is it included? Is this perhaps a mistake and a different figure S3 was meant to be included?

Actually, the comparator reaction process is complicated. Therefore, we modified Figure 6 to add careful, detailed explanations of the comparator. By the modification, Figure 6 became too large, so we separated the figure into new Figure 6, describing only the mechanism, and new Figure 7, describing only the experimental and numerical results as follows:

Figure 6. Application of pathway control to a molecular comparator for miRNA concentrations. (a) Concept of a molecular comparator of miRNA concentrations. The triangle is a symbol for a comparator element. miRNAs miR-6875-5p and miR-4634 were used for Input 1 for the comparator; miR-1246 and miR-1307-3p were used for the Input 2. The Output is the selection of the droplet division pathway, which changes depending on the difference between two initial total concentrations of miRNAs (working as inhibitor RNAs), c_{AB} and c_{AC} . This concentration comparison is achieved by the two time-delay circuits as shown in (b) and (c). (b) Pathway 1 is selected: if the Input 1 concentration is larger than the Input 2 concentration ($c_{AB} > c_{AC}$), the L_{AB}^\dagger cleavage delays longer than the L_{AC}^\dagger because more R_{AB}^\dagger causes a longer time delay of the L_{AB}^\dagger cleavage. Thus, C-droplet is divided first, and B-droplet is divided subsequently. (c) Pathway 2 is selected: if $c_{AB} < c_{AC}$, the L_{AC}^\dagger cleavage delays longer. Thus, B-droplet is divided first, and C-droplet is divided subsequently.

Figure 7. Experimental and simulation results of molecular concentration comparator. (a) Time courses of r_{div_B} (blue) and r_{div_C} (red) at varying the two normalized initial total concentrations of inhibitor RNAs \tilde{c}_{AB} and \tilde{c}_{AC} in the experiment. $\tilde{c}_{AB} = u_{R^+_{ABi}}^{\text{tot}}/u_{T^+_{ABi}}^{\text{tot}}$ and $\tilde{c}_{AC} = u_{R^+_{ACi}}^{\text{tot}}/u_{T^+_{ACi}}^{\text{tot}}$ ($i=1,2$), where the input initial total RNA concentrations are defined as $u_{R^+_{AB1}}^{\text{tot}} = [\text{miR-6875-5p}]$, $u_{R^+_{AB2}}^{\text{tot}} = [\text{miR-4634}]$, and $u_{R^+_{AB1}}^{\text{tot}} = u_{R^+_{AB2}}^{\text{tot}}$; $u_{R^+_{AC1}}^{\text{tot}} = [\text{miR-1246}]$, $u_{R^+_{AC2}}^{\text{tot}} = [\text{miR-1307-3p}]$, and $u_{R^+_{AC1}}^{\text{tot}} = u_{R^+_{AC2}}^{\text{tot}}$. The $\Delta \tilde{c}$ ($= \tilde{c}_{AB} - \tilde{c}_{AC}$) was varied at (i) 1.25, (ii)

0.50, (iii) -0.50 , (iv) -1.00 , and (v) -1.25 . RNase H concentration was fixed at $0.25 \text{ U}/\mu\text{L}$ in all experiments. The plots in conditions (i) and (v) are identical to those in Figures 5f and 5g, respectively. Error bars: standard errors of 3 observations. (b) Time difference $\Delta\tau$ at each of five RNA conditions (i)-(v) in the experiment. Error bars: standard errors of more than 4 observations. (c) Schematic of the pathway selection depending on the \tilde{c}_{AB} , \tilde{c}_{AC} , and offset concentration σ in the experiment. σ was estimated as -0.75 , which is the average of $\Delta\tilde{c}$ between conditions (iii) and (iv). (d) Time courses of r_{div_B} (blue) and r_{div_C} (red) at varying inhibitor RNA concentrations in the reaction-diffusion simulation. The $\Delta\tilde{c}$ was varied at (i) 1.25 , (ii) 0.50 , (iii) -0.50 , (iv) -1.00 , and (v) -1.25 . The hybridization rate and the strand displacement rate between T_{ABi}^\dagger and L_{AB}^\dagger were set 10 times lower than those between T_{ACi}^\dagger and L_{AC}^\dagger , respectively. Threshold parameters K_{AB} and K_{AC} were set as 0.1 and 0.9 , respectively. (e) Time difference $\Delta\tau$ at each of five RNA conditions (i)-(v) in the reaction-diffusion simulation. The hybridization rate and the strand displacement rate between T_{ABi}^\dagger and L_{AB}^\dagger were set 10 times lower than those between T_{ACi}^\dagger and L_{AC}^\dagger , respectively. $K_{AB} = 0.1$ and $K_{AC} = 0.9$ ($\sigma \neq 0$). (f) Time difference $\Delta\tau$ at each of five RNA conditions (i)-(v) in the reaction-diffusion simulation. The hybridization rate and the strand displacement rate between T_{ABi}^\dagger and L_{AB}^\dagger were the same as those between T_{ACi}^\dagger and L_{AC}^\dagger , respectively. $K_{AB} = 0.1$ and $K_{AC} = 0.1$ ($\sigma = 0$) (f).

The description of the comparator in the main text was also modified as follows:

Finally, we applied the pathway control of droplet division to a molecular computing element “comparator” of RNA concentrations. Figure 6a shows the concept of the comparator using the division pathway of the $C \cdot A \cdot B$ -droplet (details are explained below using Figures 6b and 6c). In this comparator, Input is the initial total

concentrations of miRNA sequences that are used as inhibitor RNAs in the time delay circuit. Specifically, Input 1 (c_{AB}) is the concentration of R^{\dagger}_{ABi} ($i=1,2$; miR-6875-5p and miR-4634) used in the time delay circuit for the delay of B-droplet division. Input 2 (c_{AC}) is the concentration of R^{\dagger}_{ACi} ($i=1,2$; miR-1246 and miR-1307-3p) used in the time delay circuit for the delay of C-droplet division. Output is the selection result of the division pathway depending on the differences two Inputs (c_{AB} and c_{AC}).

The details of the reaction scheme are shown in Figures 6b and 6c. If $c_{AB} > c_{AC}$ (Figure 6b), the L^{\dagger}_{AB} cleavage delays longer than the L^{\dagger}_{AC} because more R^{\dagger}_{ABi} causes a longer time delay of the L^{\dagger}_{AB} cleavage; then, C-droplet is divided first, and B-droplet is divided subsequently, which means that Pathway 1 is selected. On the other hand, if $c_{AB} < c_{AC}$ (Figure 6c), the L^{\dagger}_{AC} cleavage delays longer; then, B-droplet is divided first, and C-droplet is divided subsequently, which means that Pathway 2 is selected. Thus, the observation of the selected pathway indicates the result of the concentration comparison between Inputs, c_{AB} and c_{AC} .

We are sorry for our explanations hard to comprehend. The gray triangle is a ‘symbol of comparator’ used in general in computer science. That is, what we constructed in this study is the reactions inside the ‘triangle symbol’; i.e., the triangle element works as a molecular comparator composed of RNase H-based trigger inhibition and phase separation of droplets. The multiple reactions described in Figures 6b and 6c compose the comparator element (i.e., the ‘triangle’ element). We carefully corrected our explanations as shown in the above responses. We believe the detailed descriptions with the new way of illustrations (Figures 6b and 6c) in the revised manuscript help the understanding of the comparator mechanism. In addition, as the reviewer pointed out, since the illustration in the previous supplementary information was not so different from the illustration in the main text and the detailed explanations are described in the current main text, we removed the illustration in the supplementary information in the revised version.

We are sorry for our explanations hard to comprehend, again. The parameter ‘sigma’ is called ‘offset’ of comparator in general. For example, given x and y , ideally, one wants to compare the two values x and y ; i.e., one wants to know whether $x > y$ ($x - y > 0$) or $x < y$ ($x - y < 0$) using a comparator element. However, because of some reasons in an electric circuit, the comparator has a bias σ , which is called offset. Therefore, in general, a comparator can tell us only whether $x - y > \sigma$ or $x - y < \sigma$ (ideally, σ should be zero, but usually non-zero). In our system, the difference in the response of the division to the linker cleavage may have caused the bias. However, actually, it is difficult to pin down the causes of the bias in multistep reactions with many molecules. Here, we assumed that one of the possible contributions to the bias is the response of the division to the linker, i.e., the threshold of the division. Based on this hypothesis, we numerically investigated that offset was reproduced by changing the thresholds. As a result, it was reproduced. In the revised manuscript, we show the results more carefully: Experimental results showing the offset (Figure 7c), and numerical investigations (Figure 7d-7f) with comparison between the cases of $\sigma \neq 0$ (Figure 7e) and $\sigma = 0$ (Figure 7f). In addition, the descriptions associated with sigma in the main text was revised as follows:

(p.18)

Ideally, the sign of $\Delta\tau$ is expected to switch when $\Delta\tilde{c} = 0$ (i.e., $c_{AB} = c_{AC}$). However, the results imply that the sign switches between $-1.0 < \Delta\tilde{c} < -0.5$ (i.e., $c_{AB} \neq c_{AC}$). Here, we define an offset concentration of this molecular comparator, σ , at which the sign of $\Delta\tau$ switches, where the output of the comparator switches. Ideally, $\sigma = 0$ as shown in Figure 6a, while our molecular comparator had a non-zero offset ($\sigma \neq 0$); the σ value was estimated around -0.75 since the sign of $\Delta\tau$ switches between $-1.0 < \Delta\tilde{c} < -0.5$ (Figure 7c). Generally, regular electrical comparators also have a non-zero offset voltage because of non-ideal circuit

	properties; similarly, our molecular comparator would have had non-ideal reaction properties. We guess that $\sigma \neq 0$ would be caused probably by the difference that B-droplet division took longer than that of the C-droplet for some reasons (p.19) Figure 7f shows the ideal case that $\sigma = 0$ (explained in Figure 6), which is realized on the conditions that all the parameter values associated with L^{\dagger}_{AB} cleaving and L^{\dagger}_{AC} cleaving are equivalent (i.e., Pathway 1 and Pathway 2 are symmetric).
[4] Overall, the value of the simulations is somewhat minor. While I really appreciate the effort to be rigorous, the complexity of the model does not provide the advantages one would expect when compared to a more minimalistic, intuitive model. First, the complex model does not reproduce data in an accurate way; second, it actually loses connection with reality given the extremely high Hill coefficients adopted.	We agree that the model cannot precisely reproduce the experimental results even if one makes a detailed model in general and that the reasonability of the use of high Hill coefficient is weak. First, the value of Hill coefficient did not affect the overall results as we showed in the previous revision with supplementary numerical results; so, we chose the value only based on the reproducibility of the time courses. Next, in this study, we finally succeeded in giving one possibility to explain the comparator's offset using the model with precise reaction steps. Therefore, we believe the precise modeling would have helped to make discussion of the experimentally unobservable points and to find the way to the next-step research, although we, of course, know that it is just one possibility. In addition, we agree with the importance of the minimalistic model; thus, in future theoretical research, we would like to make such a model and compare it with the current model to understand their description ability. To clarify this point, we added the following description in the Conclusion: In addition, although our model suggested one of the possibilities of the cause of the offset concentration σ, this model is not perfect as discussed above; thus, further study on numerical modeling with experimental studies would be required.

[5] In conclusion, I think this is solid work, the results are visually appealing, and it definitely deserves publication - the approach is of interest for people working in the field of DNA condensates. I don't think the best venue for publication is a general audience journal such as this one.

We thank the reviewer for giving us many important and fruitful comments and suggestions. We believe that the revised version we made thanks to all reviewers is beneficial for a general audience as well as DNA nanotechnology researchers, since control of phase separation with chemical reactions attract attentions in many field (for example, Bergmann, A. M.; Bauermann, J.; Bartolucci, G.; Donau, C.; Stasi, M.; Holtmannspötter, A.-L.; Jülicher, F.; Weber, C. A.; Boekhoven, J. Liquid Spherical Shells Are a Non-Equilibrium Steady State of Active Droplets. Nat. Commun. 2023, 14 (1), 6552.) as well as DNA nanotechnology field.

Reviewer 4

I co-reviewed this manuscript with one of the reviewers who provided the listed reports. This is part of the Nature Communications initiative to facilitate training in peer review and to provide appropriate recognition for Early Career Researchers who co-review manuscripts.	We thank the reviewer for reviewing our manuscript again.
---	--

REVIEWERS' COMMENTS

Reviewer #3 (Remarks to the Author):

The authors have answered my points and revised the manuscript significantly! I don't have any further comment.

I now realize what the old Fig. 6 represented! Because the comparator triangle was to the right of the network, I assumed the comparator was a distinct, new part of the circuit whose output becomes the input to the network. In contrast, the network in the box to the right *is* the comparator, so the figure was supposed to convey that the circuit can be repurposed to sense concentration differences to select a pathway. I completely misunderstood. Thank you, the new figures are much clearer.

Reviewer 3

The authors have answered my points and revised the manuscript significantly! I don't have any further comment. I now realize what the old Fig. 6 represented! Because the comparator triangle was to the right of the network, I assumed the comparator was a distinct, new part of the circuit whose output becomes the input to the network. In contrast, the network in the box to the right *is* the comparator, so the figure was supposed to convey that the circuit can be repurposed to sense concentration differences to select a pathway. I completely misunderstood. Thank you, the new figures are much clearer.

We thank the reviewer for reviewing our manuscript again.